# Feynman: Knowledge-Infused Diagramming Agent for Scaling Visual Reasoning Data

## Abstract

Visual reasoning is an essential ability of state-of-the-art multi-modal AI systems. Improving these systems requires high-quality vision-language data at scale. Despite the abundance of internet image and text data, knowledge-rich and well-aligned image-text pairs are rare. In this paper, we present a scalable data generation pipeline built with our diagramming agent, Feynman. To create diagrams, Feynman first enumerates domain-specific knowledge components ("ideas") and performs code planning based on the ideas. Given the plan, Feynman translates ideas into simple declarative programs and iterates to receives feedback and visually refine diagrams. Finally, the declarative programs are rendered by the Penrose diagramming system. The optimization-based rendering of Penrose preserves the visual semantics while injecting fresh randomness into the layout, thereby producing diagrams with visual consistency and diversity. As a result, Feynman can author diagrams along with grounded captions with very little cost and time. Using Feynman, we synthesized a dataset with more than $100k$ well-aligned diagram-caption pairs. We also curate a visual-language benchmark, Diagramma, from freshly generated data. Diagramma evaluates the visual reasoning capabilities of vision-language models. We plan to release the dataset, benchmark, and the full agent pipeline as an open-source project.

## 1 Introduction

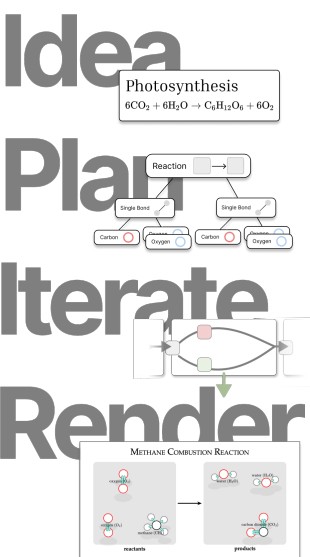

Fig. 1: The Feynman Agent

A diagram is worth ten thousand words. Humans represent knowledge visually and solve complex problems efficiently using diagrams (Tversky, 2017; Larkin & Simon, 1987). However, the current generation of multi-modal large-language models (MLLMs) such as GPT-4V (Yang et al., 2023), Gemini (Team et al., 2023) and Llama 3 (Dubey et al., 2024b) still struggle to understand, use, and generate simple visual objects that often show up in diagrams, despite tremendous progress on general multi-modal benchmarks (Yue et al., 2024). Prior work have shown MLLMs to fail rudimentary vision tests (Rahmanzadehgervi et al., 2024), perceive graph structures poorly (Li et al., 2024d), and lack compositional understanding of visual attributes, relations, and ordering (Yuksekgonul et al., 2022). The important work (Zhang et al., 2024a) specifically demonstrated their weaknesses on reasoning with abstract mathematical diagrams.

Currently, training large models relies heavily on enormous amount of data for both pre- and post-training to make progress on any capabilities (Dubey et al., 2024a; Li et al., 2024c; Tong et al., 2024), including diagram understanding (McKinzie et al., 2024). To augment training data, one general strategy is to synthesize data using state-of-the-art large-language models. Unfortunately, synthesizing vision-language data is challenging. Prevalent approaches of synthesizing vision-language data focus on the language side, such as augmenting instruction-following data from image captions (Li et al., 2022; Liu et al., 2024b; Wang et al., 2022). To synthesize images, the two main paradigms are diffusion models (Rombach et al., 2022) and graphics program synthesizers (Belouadi et al.,

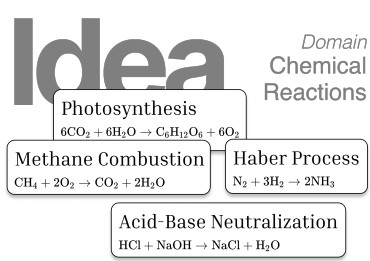
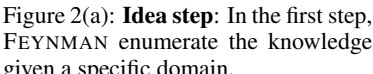

Figure 2(a): **Idea step**: In the first step, FEYNMAN enumerate the knowledge given a specific domain.

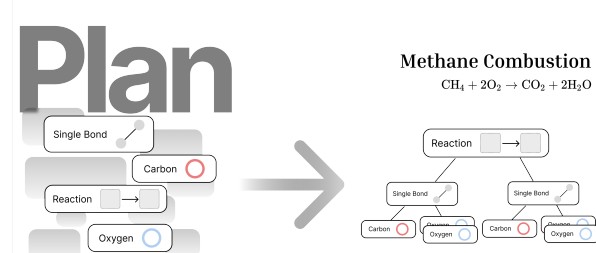

Figure 2(b): **Plan step**: Per idea, FEYNMAN then devises extract relevant elements such as chemical bonds and formulate a plan to translate them into Substance code.

2024b;c; Wu et al., 2023). The former generate raster images while the latter synthesize textual programs that produce vector graphics. Regardless of the output format, both approaches struggle to produce good diagrams consistently. Given the demand for high-quality synthetic diagrams and the limitations of current approaches, we ask the following research question:

*Can we generate synthetic diagram-language pairs at scale?*

When generating conceptual diagrams, models are tasked to perform both *knowledge elicitation* and *visual production*. When prompted to produce a diagram representing some high-level concepts, the model needs to elicit the relevant concepts (abstract knowledge, e.g., $H_2O$ has 2 hydrogen and 1 oxygen atoms), map them to visual components (visual knowledge, e.g., use ball-and-stick model to represent molecules), and organize these components in an image (visual production). However, state-of-the-art diffusion and language models struggle because they are asked to perform these steps all at once. For instance, diffusion models can produce visually pleasing images but may ignore important concepts in the diagram; language models may include the right concepts in the image but the diagram layout can be poor and illegible. In this paper, we recognize this challenge of diagram generation, and ask:

*Can we decouple knowledge elicitation and visual production in diagram synthesis?*

To address our questions, we elicit domain knowledge from LLMs and iteratively produce high-quality diagrams using a knowledge-aware diagram interface. We propose FEYNMAN, an LLM agent that scales up diagram synthesis by decoupling knowledge elicitation and visual production. FEYNMAN leverages the knowledge advantage of modern LLMs by isolating knowledge elicitation as its first step in diagram synthesis. Instead of directly producing SVG source code or a raster image, FEYNMAN produces knowledge components ("ideas"), which are then translated to their visual representations. To ensure high-quality visual production, FEYNMAN utilizes the PENROSE language, which explicitly codifies the mapping from domain-specific concepts to their visual representations (Ye et al., 2020). The resulting diagram synthesis pipeline preserves the semantics of the diagrams, and we further utilize them to generate a diverse question-answer set tailored to the generated content. Overall, our contributions include:

1. We created a diagramming agent, FEYNMAN, to author knowledge-infused diagrams and achieves remarkable yield rate in generating textbook-level diagram examples. Powered by PENROSE, FEYNMAN generates diagrams with diverse visual content.

2. With FEYNMAN, we generated 10693 knowledge-infused programs, leading to the creation of 106930 well-aligned diagram-caption pairs. This was accomplished within 1,550 million input and output tokens at a cost of under $400 with GPT-4o-mini.

3. We release a new benchmark DIAGRAMMA made of entirely fresh examples authored by FEYNMAN. We conducted a thorough quantitative evaluation of **17** MLLMs in Table 2

4. Via comprehensive ablations and analysis, we provide insights into how to build multimodal AI agents that can work in the intersection of knowledge, visual design, and code generation. We analyze the economic aspects of synthesizing large-scale scientific diagrams using our pipeline.

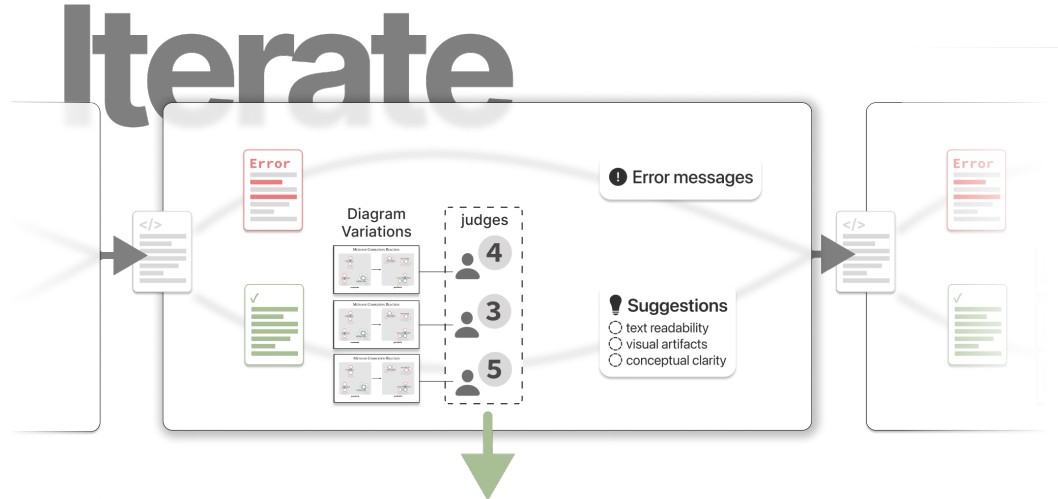

Figure 3: **Iterate Step**: At each step, FEYNMAN attempts to write PENROSE program to create a diagram. The generated program is then compiled into images and sent to a panel of visual judges (MLLMs) for critical feedback. We term this algorithm **Iterative Visual-Refine** (Algorithm 1).

Among prior work that explored similar directions, Belouadi et al. (2023) and Belouadi et al. (2024a) collected datasets of TikZ diagrams from the internet and arXiv articles to train coding agents for TikZ programs. AUTOMATIKZ (Belouadi et al., 2023) is an LLM coding agent that writes TikZ programs given text captions. However, after being trained with hundreds of thousands of TikZ diagrams scraped from arXiv LaTeX sources, AUTOMATIKZ still exhibits efficiency overhead in synthesizing scientific diagrams at scale, due to the inherent complexity of both the TikZ language and visual design. In fact, both Belouadi et al. (2023) and Belouadi et al. (2024a) requires time-consuming tree search to boost compile success rates for simple programs, making large-scale generation infeasible. In general, there is still a lack of economical and scalable solution for generating diagrams embedded with rich knowledge.

## 2 DIAGRAMMING AGENT PIPELINE

In this section, we present the workflow of our diagramming agent, FEYNMAN. FEYNMAN's diagram synthesis pipeline includes four steps: **idea**, **plan**, **iterate**, and **render**. By leveraging the knowledge capacity of LLMs and a conceptual diagramming tool, FEYNMAN can generate grounded and diverse visual representations of scientific concepts at scale. Our pipeline has the following characteristics:

1. *Knowledge scalability*: Our choice to use an LLM to provide general "knowledge-focused planning" decouples the domain knowledge elicitation and the domain-specific visual design. This choice alleviates the cost of obtaining diverse and high-quality knowledge for the generation of domain-specific diagrams (Section 2.2).

2. *Visual diversity*: The optimization-based approach of the PENROSE rendering engine provides visual diversity even given the same visual concept (Section 2.1), boosting visual diversity of the synthesized diagrams.

3. *Image-text alignment*: The programs written by FEYNMAN simply encode the conceptual ideas and relationships, from which the visuals are automatically derived. These programs resemble natural language descriptions of the concepts, enabling smooth translation between concepts and code.

In this section, we first present the background of conceptual diagramming, which is the foundation of our approach, and then introduce each step in the agentic pipeline of FEYNMAN.

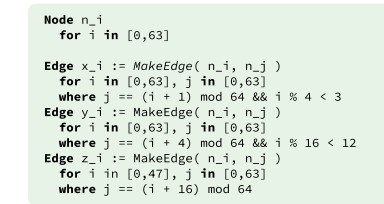
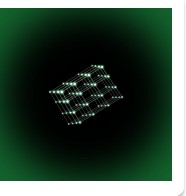
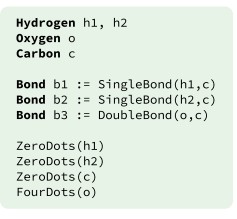
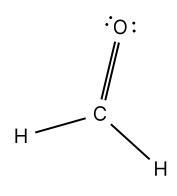

Figure 4: **Examples of conceptual diagrams and their Substance notations**: a graph where node connections form a cube (left) and the Lewis structure of the formaldehyde molecule (CH$_2$O).

## 2.1 BACKGROUND: CONCEPTUAL DIAGRAMMING

*Conceptual diagrams* refer to abstract images that visually represent "a set of ideas and their relations" (Tversky, 2017). At present, most conceptual diagrams are created by either a drawing tool like Adobe Illustrator or a low-level graphical programming language like PGF/Ti*k*Z or SVG. Using these tools is highly manual and it is extremely difficult to use them to automating diagram production (Ma'ayan & Ni et al., 2020). As a result, AI generation of diagrams through Ti*k*Z has proved challenging (Belouadi et al., 2023) (Fig. 7).

PENROSE is a diagramming tool specifically targeting conceptual diagrams. PENROSE separates the abstract concepts in the diagram (the Substance) and the visual representations of said concepts (the Style). Substance contains no low-level visual details, it is simpler and easier to generate correctly (Fig. 4). Style defines a diverse space of *diagram variations* for any given Substance. Combining Substance and Style, PENROSE samples from this space when rendering a diagram, providing FEYNMAN the ability to produce many different examples even from just one set of concepts (Fig. 6).

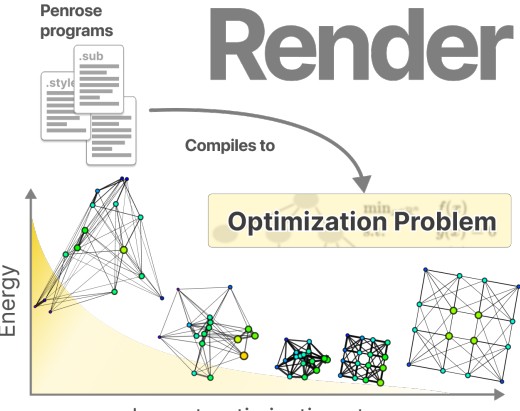

Figure 5: FEYNMAN generates programs that PENROSE compiles to generate an layout optimization problem. The PENROSE layout engine then solves the optimization problem.

To translate from concepts to visuals, Style converts the concepts and relationships from Substance into a constrained optimization problem: every concept in substance is translated to one or more shapes $\mathcal{S} = \{S_1, \ldots, S_n\}$, each with degrees of freedom $\vec{p} = (p_1, \ldots, p_m)$ such as width, height, and center. Conceptual relations among concepts are translated to *constraints* and *objectives*: constraints ensure that geometric predicates (e.g., contains, disjoint, etc.) hold true, while objectives encourage geometric relations to hold in the resulting diagram (e.g., shapes should be as far apart as possible). PENROSE encodes constraints as nonnegative penalty functions $\mathcal{P}_1, \ldots, \mathcal{P}_l : \mathbb{R}^m \to \mathbb{R}_{\geq 0}$ each of which equal 0 if and only if the constraint is satisfied. Objectives are energy terms $\mathcal{E}_1, \ldots, \mathcal{E}_k$. Overall, the PENROSE layout engine solves an optimization problem:

$$\min_{\vec{p} \in \mathbb{R}^m} \sum_{i=1}^{k} \mathcal{E}_i(\vec{p}) \quad \text{s.t.} \quad \sum_{i=1}^{l} \mathcal{P}_i(\vec{p}) = 0. \tag{1}$$

PENROSE employs an exterior point method (Hiroshi & Tanabe, 2010) to pose this problem as a sequence of unconstrained optimization problems, where constraints are iteratively stiffened over layout steps:

$$\min_{\vec{p} \in \mathbb{R}^m} \sum_{i=1}^{k} \mathcal{E}_i(\vec{p}) + c_n \sum_{i=1}^{l} \mathcal{P}_i^2(\vec{p}), \quad n = 0, 1, 2, \cdots \tag{2}$$

PENROSE solve this layout problem by running L-BFGS with line search (Lewis & Overton, 2009).

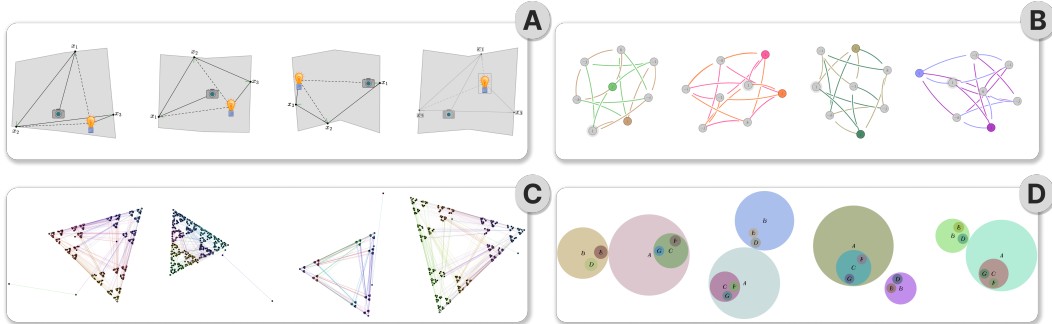

Figure 6: **Diverse visual layouts of PENROSE diagram variations:** using the same Substance, PENROSE can produce diagram variations while preserving the semantics, by sampling random initial values for shapes, colors, and other numerical quantities in the diagram. We show 4 random seed for 4 Substance programs for (A) ray-tracing diagrams, (B) Cayley graphs, (C) Chaos game as a Sierpinski triangle, and (D) Euler diagrams for sets.

## 2.2 KNOWLEDGE PLANNING: ENUMERATING THE DOMAIN KNOWLEDGE WITH AN LLM

State-of-the-art LLMs learn vast knowledge during their large-scale pretraining. For example, GPT-4o attains 53% accuracy in the GPQA benchmark (Rein et al., 2023), close to the 57% achieved by human experts who have or are pursuing Ph.D. degrees. We leverage this large capacity of knowledge of LLM by designing domain-specific prompts to ask an LLM to enumerate pieces of knowledge ("ideas") related to the selected domain. The prompts are designed to encourage LLM to perform creative knowledge enumeration. For example, LLM is given the question "Enumerate $N$ chemical reactions that are pedagogical and important." in the chemical-reactions domain (see examples of full domain planning prompts in Appendix B.2). We feed the LLM's response to the coding agent, FEYNMAN, to program the concepts into diagrams. In cases where we can't parse the output format, we try multiple rounds until we reach a maximum number of rounds.

## 2.3 DIAGRAMMING CODE PLANNING: REASONING FOR CONCEPTUAL DIAGRAMMING

In this stage, the FEYNMAN agent generates a coding plan for each of proposed knowledge components. The agent first attempts to organize the knowledge components into visual concepts, aiming to prepare for translation into PENROSE code. To make the FEYNMAN agent aware of the PENROSE syntax, we provide the official PENROSE documentation in the prompt, akin to Wu et al. (2024a). Moreover, for each domain, we provide a few in-context examples for LLM to learn the syntax for that specific domain. We then instruct FEYNMAN to plan the visual elements that are described in each knowledge component. This involves listing important steps to write a Substance program that corresponds to the sampled knowledge component (see full code planning prompts in Appendix B.4). Note that *we do not instruct the model to write runnable code in this step*, which is an explicit design choice. In Section 4, we show that they serve as crucial foundations for successful and diverse code generation.

## 2.4 ITERATIVE VISUAL-REFINE WITH A PANEL OF VISUAL JUDGES

After the planning stage, FEYNMAN translates the plan into compilable and correct PENROSE programs. We found in our preliminary experiments that even though LLMs such as GPT-4o fail to write the correct program in their first attempt, they can improve if given suitable suggestions in multi-round conversations. This is close to the **interactive self-refine** approach (Madaan et al., 2024), which improves LLM output quality via multi-round self-reflection. The refinement of diagrams, which involves visual judgments, posed different challenges. While the prior work adopt tree search to perform refinement (Belouadi et al., 2023; 2024a), their refinement process primarily aims to achieve better compilation success rate. Without visual judgments and feedback, the generated visual artifacts might include incorrect representation of knowledge.

To address this challenge, FEYNMAN utilizes a panel of visual judges to provide the visual feedback, which we term it as **Iterative Visual-Refine**. The process is illustrated in Figure 3. In the first round, FEYNMAN receives the plans from the previous planning steps and attempts to generate the first code sample. If the code is successfully compiled into a diagram, then FEYNMAN run PENROSE to generate variations of this diagram and send them to a panel of *visual judges* to assess its quality. Each visual judge is a vision-language model asked to provide critical feedback to the diagramming agent based on a set of *criteria* on various aspects of diagram qualities. To keep the task simple, we prompt the judges to answer in boolean values, which are then collected and aggregated. For cost-saving purposes, we set a threshold of scores above which we accepts the output as a valid diagram. We provide an aggregated feedback message and move to the next iteration if the program generated arrives at any of these states: a) cannot be extracted from the LLM response; b) fail to compile to diagrams or c) receive scores below the set threshold. The formal algorithm is presented in Algorithm 1.

**De-duplication.** LLM sometimes duplicate their responses given similar prompts, resulting in a lack of diversity of knowledge in diagrams. Specifically, we focus on de-duplicating the Substance code, which PENROSE uses to synthesize diagrams. We use a statement-wise Levenshtein distance (Levenshtein, 1966) to filter programs that contain too many duplicate statements. The detail is presented in Appendix B.6.

---

**Algorithm 1** Iterative Visual-Refine

**Require:** Prompt $S$, Maximum iterations $N_{\max}$, Quality threshold $\theta$, Number of Judges $K$
1: **Initial Message:** $S_0 \leftarrow S$
2: $n \leftarrow 0$ {Initialize iteration counter}
3: **while** $n \leq N_{\max}$ **do**
4:     Agent response $R \leftarrow$ FEYNMAN$(S)$
5:     Parse $R$ into PENROSE program $C$
6:     **if** Parsing returns no program **then**
7:         **continue**
8:     Compile $C$ to diagrams $d_k, \forall k \in [K]$
9:     **if** Compilation Failure **then**
10:         **Error:** $e \leftarrow$ error traceback
11:         **Message:** $S \leftarrow e$
12:         **continue**
13:     **for** each $k \in [K]$ **do**
14:         **Scoring:** $s^{(k)} \leftarrow V_k(d_k)$
15:         **Suggestions:** $S^{(k)} \leftarrow V_k(d_k)$
16:     **Average scores:** $s_{n+1} \leftarrow \mathsf{Avg}(s^{(k)})$
17:     **Average suggestions:** $S_{n+1} \leftarrow \mathsf{Avg}(S^{(k)})$
18:     **if** $s_{n+1} \geq \theta$ **then**
19:         **Exit the algorithm, returns** $C$
20:     $n \leftarrow n + 1$
21: **return** $C$

---

### 2.5 GROUNDED QUESTION-ANSWER PAIR GENERATION

For each Substance code FEYNMAN generated, we generate grounded captions from their code programs, while diversifying its visual representation through the PENROSE optimization-based layout engine. We created a multiple-choice question-answering (QA) data generation pipeline. First, we generate image captions by translating the concepts and relations in Substance to natural language. Then, we prompt an LLM to select one of five visual reasoning skill categories (Appendix B.7) for the problem. To ensure problem diversity and quality, we ask the model to provide rationales before QA generation in a chain-of-thought fashion. Finally, after a QA pair is generated, the model self-verify by checking if the question can be answered without an image and if the answer is correct given both image and question.

### 2.6 BENCHMARK CURATION: THE DIAGRAMMA BENCHMARK

We curated DIAGRAMMA, a visual reasoning benchmark using diagrams FEYNMAN synthesized. These diagrams are completely unseen and do not exist on the internet. DIAGRAMMA is a scientific benchmark that contains **1,058** multiple choice questions of visual understanding and reasoning. We manually filtered a synthetic test dataset generated by FEYNMAN to create DIAGRAMMA. We went through a meticulous filtering process to ensure that the selected images had accurate labels, correct knowledge representation, and sufficiently challenging questions. As shown in Table 1, DIAGRAMMA contains 6 subjects, each of which contains multiple subdomains e.g., sorting algorithms in computer science. Diagram counts for all subjects are listed in Table 2.

Figure 7: **Comparison of different diagramming approaches**: We select the best image out of three attempts using Flux-Pro and AUTOMATIKZ, compared with those generated by our agent. We provide them with the same prompt for each column. Flux-Pro produces visually diverse diagrams, however none of them contains legible text that matches the intent. AutomaTikZ, on the other hand, only successfully compiled for 3 out of 5 of all prompts.

## 3 EXPERIMENTS

### 3.1 SCALING DIAGRAM-CAPTION PAIRS

With the FEYNMAN agent, we conduct a preliminary dataset scaling experiment. By effectively enumerating knowledge in each subject and their subdomains, FEYNMAN produced 10693 unique Substance programs, representing diverse conceptual relationships. We used a total of 1470.4 million input tokens and 46.6 million output tokens on GPT-4o-mini to generate these Substance programs. Each program is further independently rendered by the optimization-based layout engine to produce 10 unique variations, resulting in overall 106930 diagrams. See Figure 6 for examples of diagram variations and their visual diversity. For each rendered variation, we further produce a caption according to both the image and the corresponding Substance program. The scalability of the pipeline is tested in this experiment. In Appendix D.3, we provide more detailed studies to assess the knowledge and visual diversity of generated images.

| Source metadata | |
| --- | --- |
| Subdomain | 108 |
| Substance | 1058 |

| Subjects | |
| --- | --- |
| Math | 401 (37.9%) |
| CS | 342 (32.3%) |
| Science | 241 (22.8%) |
| Chart | 30 (2.8%) |
| Common Sense | 22 (2.1%) |
| Statistics | 22 (2.1%) |
| Unique Style | 52 |
| Unique Domain | 38 |

Table 1: Metadata of DIAGRAMMA.

### 3.2 DIAGRAMMA EVALUATION

We evaluated DIAGRAMMA on state-of-the-art open- and closed-source MLLMs. Evaluations were conducted in a zero-shot setting, with a uniform template provided to each model. Detailed evaluation setup can be found in Appendix C.

**Results.** We present the evaluation results of DIAGRAMMA for a set of 17 models in Table 2. We observe three pieces of evidence validating our benchmark curation in the results: 1) as the sizes of models increase, their accuracy consistently improves, confirming the validity of our data for measuring the capabilities of MLLMs; 2) the computer science subject, which primarily involves graph

| Model Name | All | Math | CS | Science | Chart | Commonsense | Statistics |
|---|---|---|---|---|---|---|---|
| **Claude-3.5-Sonnet** | **59.64** | **64.59** | 42.98 | **74.69** | **53.33** | **77.27** | **54.55** |
| GPT-4o (OpenAI (2024a)) | 57.28 | 63.09 | **50.58** | 60.17 | **53.33** | 50.00 | 36.36 |
| Claude3-Opus (Anthropic (2024)) | 49.15 | 54.11 | 40.35 | 55.60 | 33.33 | 45.45 | 50.00 |
| Gemini-1.5-Flash (Reid et al. (2024)) | 47.54 | 55.36 | 40.06 | 45.64 | 50.00 | 54.55 | 31.82 |
| Claude3-Sonnet (Anthropic (2024)) | 47.54 | 50.62 | 38.01 | 58.92 | 46.67 | 36.36 | 27.27 |
| GPT-4o-mini (OpenAI (2024a)) | 44.42 | 47.63 | 36.55 | 53.53 | 26.67 | 45.45 | 31.82 |
| Gemini-1.5-pro (Reid et al. (2024)) | 44.23 | 49.13 | 41.23 | 42.74 | 30.00 | 59.09 | 22.73 |
| Claude3-Haiku (Anthropic (2024)) | 42.53 | 46.63 | 31.58 | 54.77 | 30.00 | 36.36 | 27.27 |
| Qwen2-VL-72B* (Yang et al. (2024a)) | 50.85 | 59.10 | 42.69 | 51.45 | 46.67 | 31.82 | 45.45 |
| LLama3.2-VL-90B* (Dubey et al. (2024b)) | 46.88 | 50.37 | 41.23 | 53.53 | 26.67 | 40.91 | 31.82 |
| LLama3.2-VL-11B (Dubey et al. (2024b)) | 46.22 | 48.38 | 40.06 | 56.02 | 30.00 | 50.00 | 13.64 |
| Pixtral-12b* | 44.71 | 50.62 | 40.06 | 44.81 | 43.33 | 36.36 | 18.18 |
| LLava-OneVision-Qwen2-7b (Li et al. (2024b)) | 42.91 | 48.13 | 35.38 | 44.40 | **53.33** | 45.45 | 31.82 |
| Qwen2-vl-7B (Yang et al. (2024a)) | 42.16 | 48.63 | 35.38 | 40.66 | 43.33 | 45.45 | 40.91 |
| InternVL2-8B (Chen et al. (2024)) | 41.02 | 47.38 | 35.09 | 39.83 | 40.00 | 40.91 | 31.82 |
| Phi-VL-3.5 (Abdin et al. (2024)) | 38.19 | 42.64 | 33.04 | 40.66 | 30.00 | 40.91 | 18.18 |
| Minicpm-2.6 (Yao et al. (2024)) | 35.44 | 41.65 | 29.82 | 36.51 | 26.67 | 27.27 | 18.18 |

Table 2: Accuracy results of DIAGRAMMA on state-of-the-art MLLMs on **1058** samples. Models marked with (*) are evaluated through the OpenRouter API. **Claude-3.5-Sonnet** achieved the highest overall accuracy, with notable performance on science and commonsense diagrams.

reasoning, remains the most difficult for current models, which corroborates with the observations by Li et al. (2024d); Rahmanzadehgervi et al. (2024). An notable observation is that Gemini-1.5 Flash, which is considerably cheaper than Gemini-1.5 Pro outperformed Gemini-1.5 Pro on DIAGRAMMA, a trend also seen in the reasoning category in LIVEBENCH (White et al., 2024). We conjecture this correlation is attributed to that DIAGRAMMA share the same "freshness" as the reasoning questions in LIVEBENCH. We find that Gemini 1.5 Pro declined to answer over 100 questions, contributing to a more than 10% drop in accuracy, which might indicates high rejection rates of answering out-of-distribution questions. In Appendix C.4, we present a qualitative analysis of how most MLLMs struggled in reasoning.

## 4 ANALYSIS

### 4.1 BASELINE COMPARISON

We show in Figure 7 a preliminary comparison study on 5 attempts to generate diagrams corresponding to specific prompts. We compare FEYNMAN to two competitors AUTOMATIKZ and diffusion model FLUX-Pro (fal.ai, 2024) as baselines. Each attempt of baseline was ran 3 times and we select the best result. The caption is provided in Appendix E.2. The diffusion model FLUX-Pro has difficulty generating clean scientific diagrams that conveys the concepts, but instead hallucinates many low-level details not mentioned in the caption. AUTOMATIKZ (Belouadi et al., 2023), which trained a Llama-2 model to generate TikZ code, fails to produce correct TikZ programs that match the captions. We hypothesize that their failure in our setting if because 1) complex diagrams require great precision to draw than natural images, and 2) TikZ programs cannot separate knowledge organization and visual production, which made the code synthesis task overly difficult. We provide further explorations of using o1-mini and o1-preview (OpenAI, 2024b) to generate TikZ code in Appendix E.2. However, even when powerful LLMs like o1-preview can write TikZ code roughly correctly, TikZ still makes it hard to diversify the layout.

### 4.2 PRODUCTION-TO-SCALE ANALYSIS

We conducted an ablation experiment shown in Fig. 8 highlights the scalability of FEYNMAN. In knowledge-dense domains, the agent scales its knowledge linearly as the number of tokens increases. Even in knowledge-sparse domains, we observe an upward trend in token generation, though at a reduced rate. This demonstrates the robustness of FEYNMAN 's performance across varying levels of domain.

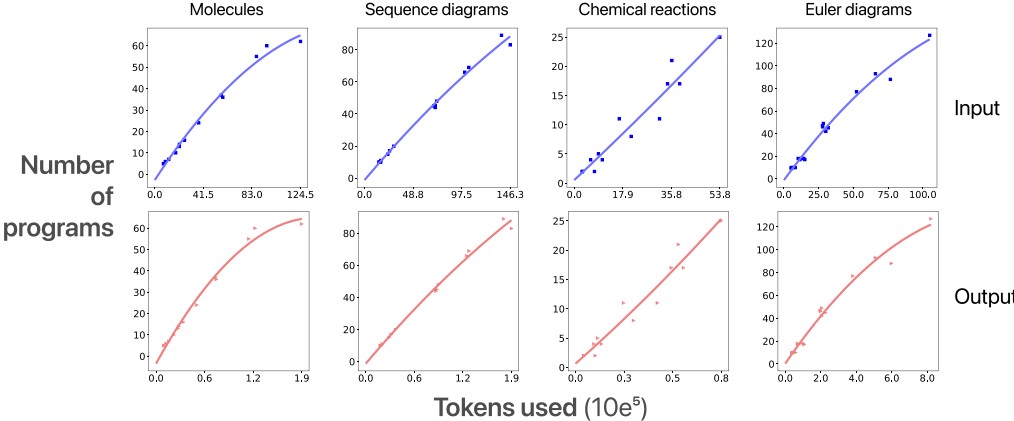

Figure 8: Scaling behavior of input/output tokens versus generated image samples across four sub-domains. The figure illustrates two distinct trends: linear scaling in knowledge-dense domains, and decaying trends in knowledge-sparse domains. These trends highlight the impact of domain knowledge diversity and scalability on the performance of FEYNMAN.

The results of our production-to-scale experiment on four selected subdomains are illustrated in Fig. 8. The plots show the total input/output tokens versus the number of generated images after de-duplication. Across the domains, we observe two distinct scaling patterns: a linear trend and a decaying trend. The linear trend suggests the potential for further scaling within a specific domain as the number of tokens increases further. In contrast, the decaying trend indicates diminishing returns in the generation of images as token counts continue to rise. We attribute these differing trends to two main factors: (1) the base model's knowledge within the specific domain, and (2) the scalability of the domain for diagramming. Additionally, domains can be classified as either knowledge-dense or knowledge-sparse: for instance, enumerating knowledge about chemical reactions is likely easier than for a domain composed solely of complementary triangles.

### 4.3 ABLATIONS FOR AGENT WORKFLOW

To provide quantitative insights into the effectiveness of FEYNMAN, we evaluate our pipeline on two metrics: the **final yield rate** is percentage of successfully complied images after de-duplication; the **visual judge scores** is an average of rule-based critic score given by MLLM judges for de-duplicated images at the end of the generation. We perform an ablation study on 10 subdomains that encompass wide range of knowledge. As shown in Table 3, we ablate key components of FEYNMAN: (1) explicit knowledge planing (**KP**, Section 2.2), (2) explicit code planning (**CP**, Section 2.3), and (3) early stop mechanism based on judge scores (**S**, Section 2.4).

We discuss three notable findings from our ablation results. First, the pipeline with all components (**KP + CP + S**) achieves the best average judge score. Second, code planning (**CP**), together with use score to early stop, helps the generation pipeline end in much lower number of rollout rounds. Finally, when knowledge planning (**KP**) is present, the gap between compiled success rate and final yield rate is very low. The observations above suggest that knowledge planning (**KP**) is essential for generating diverse scientific diagrams. Additionally, code planning (**CP**) improves the scalability of the data generation pipeline. When combined with early-stop, **CP** helps generate better-quality figures with fewer iterations and reduce the overall cost. Combing both steps achieves our goal of generating *scalable* and *diverse* diagrams.

## 5 RELATED WORK

**Multi-modal LLMs and agents**  Vision-language models (Alayrac et al., 2022; Yang et al., 2023; Li et al., 2023a; Zhu et al., 2023) gained remarkable capability of following instructions through visual instruction tuning (Li et al., 2022; Liu et al., 2024b;a). This capability enabled wide range of applications, such as visual reasoning(Yue et al., 2024; Lu et al., 2023), and interact with human as

| Ablations | Avg | PR | LT | SP | NR | CR | Compile % | Yield % | Rounds |
|---|---|---|---|---|---|---|---|---|---|
| KP + CP + S | **65.4** | **69.7** | 30.9 | **60.4** | 91.8 | **74.2** | 82.5 | 82.5 | 2.63 |
| CP + S | 65.0 | 65.9 | 32.1 | 59.9 | **95.0** | 72.2 | **97.0** | 87.5 | **2.44** |
| KP + S | 62.8 | 63.9 | **35.6** | 56.8 | 91.3 | 66.2 | 87.0 | 86.5 | 6.69 |
| S | 63.2 | 64.3 | 31.5 | 57.6 | 90.6 | 71.8 | 96.5 | 84.0 | 6.29 |
| KP + CP | 61.5 | 67.9 | 25.9 | 52.1 | 89.7 | 71.9 | 81.5 | 81.5 | 8.00 |
| CP | 61.8 | 69.3 | 23.6 | 52.5 | 92.3 | 71.1 | 96.5 | **92.5** | 8.00 |
| KP | 63.9 | 66.6 | 32.1 | 59.5 | 91.2 | 70.2 | 87.5 | 87.5 | 8.00 |

Table 3: The table shows a break down of critic scores in five categories (see Appendix D.1, **PR**: proper element relationship; **LT**: legible text; **NR**: non-redundancy; **CR**: correct representation; **SP**: simplicity), diagram compilation rate (**Compile %**), final yield rate (**Yield %**), and total number of rounds in the iterate step (**Rounds**). The combination of explicit knowledge planning (**KP**), code planning (**CP**), and early stop based on scoring results (**S**) received the best judge critic score.

visual chatbots OpenAI (2024a). Meanwhile, agents built with LLMs can interact with environments (Wang et al., 2024c) to play games, perform web navigation, and write computer programs (Wang et al., 2023; Yao et al., 2022; Romera-Paredes et al., 2024; Yang et al., 2024c; Xia et al., 2024; Wu et al., 2024b). MLLM agents have more perception modalities and are more grounded in real-world scenarios (Hong et al., 2024; Sun et al., 2022; Li et al., 2024a; Wang et al., 2024a; Bonatti et al., 2024; Koh et al., 2024; Li et al., 2024e). Agents are also important collectors of data, but the efficiency of data collection depends on the domain of choice (Putta et al., 2024).

**Synthetic data generation** The success of large AI models depends primarily on the scaling law of model size and training data (Kaplan et al., 2020; Hoffmann et al., 2022), which stimulated efforts to curate datasets (Gao et al., 2020; Soboleva et al., 2023; Li et al., 2024c). For domains where data collection is expensive, synthesizing data has become the dominant approach (Haluptzok et al., 2022; Zelikman et al., 2022; Yang et al., 2024b; Li et al., 2023b; Wang et al., 2022; Peng et al., 2023). Synthetic data have long existed in the vision domain (Little & Verri, 1989). For multi-modal AI models, teaching the model to harness their visual reasoning capabilities also relies on synthetic data (Li et al., 2022; Liu et al., 2024b;a). One work that tackled similar problems to ours is by Zhang et al. (2024b), who synthesized charts and figures via LLM knowledge and program synthesis, but their approach is limited by the tool of choice and lack of agentic ability such as iterative refinement.

**Vision-language benchmarks** Sustainable progress of AI research relies on the continuous development of benchmarks to measure the capabilities of AI systems. Benchmarks like HellaSwag (Zellers et al., 2019; Hendrycks et al., 2020; Cobbe et al., 2021; Zheng et al., 2023) contributed significantly to the progress measurement of building state-of-the-art LLMs. Vision-language benchmarks serve the same role for the visual understanding and reasoning abilities of MLLMs. For example, benchmarks like VQA-v2 (Antol et al., 2015), GQA (Hudson & Manning, 2019) and MMMU (Yue et al., 2024) measure the visual knowledge of MLLMs comprehensively. Other benchmarks like (Methani et al., 2020; Lu et al., 2021b;a; Masry et al., 2022; Lu et al., 2024; Wang et al., 2024b) measure domain-specific capabilities like chart understanding and math visual reasoning.

## 6 CONCLUSION

In this paper, we presented FEYNMAN, a diagramming agent that authors conceptual diagrams at scale. FEYNMAN decouples knowledge elicitation from visual production of diagrams to achieve scalability of diagram synthesis. Grounded by a knowledge-infused diagramming language, FEYNMAN produces text-image pairs to scale up the synthetic data across multiple subjects, such as computer science and mathematics. We conducted systematic ablation of key design choices for FEYNMAN, and showed the production-to-scale curves to demonstrate the scalability of our pipeline. Additionally, we released a new benchmark DIAGRAMMA with question-answer pairs generated by FEYNMAN and curated to ensure correctness, further contributing to research in diagram-based reasoning.

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

## A LIMITATION AND FUTURE WORK

**Dependency on Implicit Knowledge:** FEYNMAN relies heavily on the implicit knowledge embedded in large language models (LLMs) during knowledge elicitation. This reliance can reduce the model's effectiveness in domains where LLMs have incomplete or biased knowledge, resulting in less diverse outputs. A key area for future work is improving how knowledge is elicited from LLMs. For example, this may involve integrating a Retrieval-Augmented Generation (RAG) pipeline to supplement LLMs with external, domain-specific information.

**Limited Diagram Style Variation:** While the generated diagrams exhibit layout diversity, control over stylistic elements, such as color schemes or visual aesthetics, is limited to the default capabilities of the PENROSE language. To address this, future efforts will focus on systematically varying Style and Domain programs in PENROSE, enabling more flexible and customizable diagram generation.

## B DETAILED PIPELINE FOR FEYNMAN

### B.1 PIPELINE CONFIGURATION

Below we create a list of hyperparameters used in FEYNMAN. An ablation study is done in Section 4 on some key hyperparameters, labeled with (*).

- **Planning LLM**: the model used during Knowledge Planning section
- **Coding LLM**: the model used during Code Planning and Generation
- **Number of Rounds**: The number of iterative improvement rounds per sample
- **Critic MLLMs**: a list of MLLM candidate used to judge the image at the end of each rollout rounds and final generation pipeline
- **Use Knowledge Planning\***: A flag to note whether to explicitly conduct one turn of knowledge planning conversation
- **Use Code Planning\***: A flag to note whether to explicitly conduct one turn of code planning conversation
- **Use Scores to Early Stop\***: A flag to note whether to use critic MLLM judge score to early exit rollout. If the flag is set to false, the code model only receives MLLM feedback.

The default configuration for FEYNMAN is to use GPT-4o as the planning model and GPT-4o-mini as coding LLM. Our MLLM candidates are selected from GPT-4o-mini, Claude-3.5-sonnet, and Gemini-1.5-Pro. The number of rollout rounds are set to 8, with all flags set to true.

918
919
920
921
922
923
924
925
926
927
928
929
930
931
932
933
934
935
936
937
938
939
940
941
942
943
944
945
946
947
948
949
950
951
952
953
954
955
956
957
958
959
960
961
962
963
964
965
966
967
968
969
970
971

## B.2 Prompt for Knowledge Planning

In this section, we provide knowledge planning prompts in some example domain. These prompts aim to encourage LLM's to elicit its pretrain knowledge to think of creative scenarios in a given domain, specially for elements that can be altered through substance code.

---

**Geometry:**

As a geometry teacher, think of various ways to draw geometric shapes and their constructions with clear labeling. Your goal is to help students understand the properties and relationships of geometric figures through detailed and thoughtfully designed diagrams. Outline a variety of diagrams that could be drawn in this domain. Guidelines:

1. Concept Focus:
- Shape Types: Utilize various shapes like triangles, quadrilaterals, rectangles, circles, and angles to demonstrate different geometric principles.
- Geometric Constructions: Show constructions such as bisectors, perpendicular bisectors, midpoints, and angle formations to illustrate fundamental concepts.
- Relationships and Properties: Highlight geometric relationships and properties such as parallelism, equal lengths, and angle measures.

2. Planning Elements:
- Diagram Layout: Decide on the arrangement of shapes and lines to clearly show their relationships and constructions. Consider layouts that logically progress through the steps of construction.
- Labeling: Plan to consistently and clearly label all points, lines, and angles to enhance understanding. Ensure labels do not clutter the diagram and are easily readable.

---

**Word Cloud:**

As a high school English teacher, generate engaging word clouds to help students visualize key concepts in literature. Word clouds should highlight word frequency and importance to aid understanding of themes, vocabulary, and literary devices.

Guidelines:

1. Focus on concepts like themes, important vocabulary, literary devices, and text analysis.
2. Use texts appropriate for high school students, exclude common stop words, and design for readability and appeal.
3. Ensure significant words are prominent and visuals accurately reflect word emphasis.
4. Examples:
- Word cloud of common words in a Shakespearean soliloquy.
- Word cloud highlighting key vocabulary from a novel chapter.
- Word cloud showcasing sensory words in a descriptive essay.
5. Keep word clouds simple, use them to prompt discussions, and include brief annotations or questions to encourage critical thinking.

**Matrix Operation:**
As a mathematics teacher focusing on matrix operations, design a variety of diagrams to illustrate fundamental matrix and vector operations. Your goal is to help students understand the essential principles and applications of matrix operations through thoughtfully designed diagrams. Outline a range of diagrams that could be drawn in this domain. Guidelines:

1. Concept Focus:
- Basic Elements: Represent scalars, vectors, and matrices to demonstrate foundational concepts.
- Matrix and Vector Operations: Highlight important operations such as transposition, scalar multiplication, matrix multiplication, vector addition, and element-wise operations.
- Applications: Optionally illustrate applications of matrix operations in solving linear equations, transformations, and other practical scenarios.

2. Planning Elements:
- Diagram Layout: Arrange matrices, vectors, and scalars clearly to show their relationships and operations. Consider using simple, clean layouts to avoid confusion.
- Labeling: Clearly label all elements (matrices, vectors, scalars) and their components for easy identification. Use consistent and concise labeling throughout the diagrams.

3. Diverse Diagrams Examples:
- Transpose of a Matrix: Draw a matrix and its transposed version to illustrate the concept of matrix transposition.
- Scalar Multiplication: Show examples of scalar multiplication with matrices and vectors, demonstrating how each element is scaled.

4. Educational Focus:
- Clarity: Ensure each diagram is easy to interpret and effectively clarifies matrix operations concepts for students. Strive for simplicity and avoid unnecessary complexity in the diagrams.

Use these guidelines to outline a series of well-organized and informative diagrams that will effectively aid in teaching the principles and applications of matrix operations.

**Chemistry Structural Formula:**

As a high school teacher in chemical synthesis and reaction design, I need your help to generate novel and potentially useful chemical reactions. Please follow these guidelines:

1. Consider various types of organic and inorganic reactions, including but not limited to:
- Carbon-carbon bond formations
- Oxidation and reduction reactions
- Substitution reactions

2. Take into account different reaction conditions such as:
- Temperature ranges
- Pressure conditions
- Solvents
- Catalysts
- pH levels

3. For each proposed reaction:
- Provide the balanced chemical equation
- Suggest possible reaction mechanisms
- Describe the expected products and any significant side products
- Explain the potential significance or applications of the reaction

Please write down the formula of the reactants and products in the chemical reaction.

## B.3 PROMPT TO SCALE KNOWLEDGE PLANNING PROMPTS

The prompt includes few-shot examples across various related domains, each with a corresponding PENROSE domain code and a manually crafted prompt. The domain code is provided as it best captures key knowledge elements essential to image construction in PENROSE, making it the most relevant information to include in a knowledge planning prompt.

---

Create a prompt that encourages the model to generate creative scenarios within a specific domain. The prompt should guide the model to identify scalable knowledge elements in that domain and ensure the output is clear and easy to understand. Here are a few examples:

**Example Domain 1**: {example domain name 1}

**Example Domain Code 1**: {example Penrose domain code 1}

**Example Domain Prompt 1**: {example domain knowledge planning prompt 1}

**Example Domain 2**: {example domain name 2}

**Example Domain Code 2**: {example Penrose domain code 2}

**Example Domain Prompt 2**: {example domain knowledge planning prompt 2}

**Example Domain 3**: {example domain name 3}

**Example Domain Code 3**: {example Penrose domain code 3}

**Example Domain Prompt 3**: {example domain knowledge planning prompt 3}

Now generated knowledge planning prompt for the given domain name and code

**Domain name**: {domain name}

**Domain code**: {domain code}

**Domain Prompt**:

---

B.4   PROMPT FOR CODE PLANNING

---

{domain_instructions}

Now I give you the domain definition, and the corresponding domain code. You can also refer to the documentation of the domain and substance in penrose for deeper understanding.

Domain documentation: {documentation_content}

Now here is an example of the substance code in this domain for{domain_name}:

substance:{substance_code_shot_content}

Given your planning above, please plan a few important steps for generating the substance code for {idx}-th example given the domain and style.

Review the plan provided and ensure a clear understanding of each step. Before generating the code, think through the following:

1. Enlist the components of the particular example.
2. How does each step translate into the penrose substance code?

Write down the reasoning and the steps you will take, especially the elements defined in the domain you will put on the diagram and the relations between them.

---

The prompt includes several key Python formatting elements:

- Domain Instructions: Hand-crafted instructions specific to code generation within a domain. For example, in the geometry domain, these instructions may guide the model to use concise labeling for each shape.

- Documentation content: Relevant PENROSE documentation, primarily covering syntax for domain, substance, and style code.

- Substance code shot content: Example compilable substance codes from the same domain, sourced either from the official PENROSE repository[1] or previous successful iterations.

---

[1] https://github.com/penrose/penrose/tree/main/packages/examples/src

### B.5 MULTI-JUDGE CRITICS

For each round, we have a number of MLLM judges to assess the quality of compiled images based on pre-defined criterion. Our candidate judges are `GPT-4o`, `Claude-3.5-sonnet`, `Gemini-Pro-1.5`. Each judge is randomly selected from candidate pool with a different random seed. The criterions are listed as follows:

- Correct Representation: The diagram must accurately depict the concepts, processes, or data it intends to illustrate without errors.

- Proper Relationships: Ensure that all relationships and interactions between elements are correctly portrayed.

- Legible Text: If there is any labels, legends, and annotations, then they should be easily readable. Long labels and annotations should be avoided.

- Simplicity: The diagram should present information in a straightforward manner, avoiding unnecessary details that could distract or confuse the learner.

- Cultural Sensitivity: Avoids symbols or imagery that might be culturally insensitive or misunderstood by the target audience.

- Organized Structure: Elements should be arranged logically to guide the reader's eye through the information seamlessly.

- No Unnecessary Repetition: Ensures that each element serves a purpose without redundant information that could clutter the diagram.

> You are given one diagram generated by the user via graphics programs. The creative intent of the diagram is:
>
> {diagram_intent}
>
> Do you think the diagram preserves the creative intent well? Please evaluate the validity of the diagram based on the following criteria, and provide a short suggestion for improvement at the end:
>
> {criterion}
>
> Format your answer as follows:
>
> Comment: [YOUR COMMENT]
>
> Correct Representation Criterion Satisfied: If you think the criterion is satisfied, say yes <GOOD>, if not, say no <BAD>. Proper Relationships Criterion Satisfied: If you think the criterion is satisfied, say yes <GOOD>, if not, say no <BAD>. Legible Text Criterion Satisfied: If you think the criterion is satisfied, say yes <GOOD>, if not, say no <BAD>. Simplicity Criterion Satisfied: If you think the criterion is satisfied, say yes <GOOD>, if not, say no <BAD>. Cultural Sensitivity Criterion Satisfied: If you think the criterion is satisfied, say yes <GOOD>, if not, say no <BAD>. Organized Structure Criterion Satisfied: If you think the criterion is satisfied, say yes <GOOD>, if not, say no <BAD>. No Unnecessary Repetition Criterion Satisfied: If you think the criterion is satisfied, say yes <GOOD>, if not, say no <BAD>.
>
> Suggestions: [YOUR SUGGESTIONS]

The format instruction allows us to use the following regex code to parse the the binary score (1 for good, 0 for bad). A total score is calculated based on average score for each judge.

```
score_pattern = r"(?i)<(good|bad)>"
suggestion_pattern = r"(?i)\bsuggestion[s]?:\s*(.*)"
```

## B.6 DE-DUPLICATION DETAILS

During de-duplication, we focus on filtering similar substance codes within the same domain. Borrowing the idea from Levenshtein distance, for each new substance code $s$ and existing set of final substance $\mathbf{S}$, we determine whether to add $s$ to $\mathbf{S}$ based on the following algorithm

---

**Algorithm 2** Determine whether to add sample $s$ to set $\mathbf{S}$ based on threshold $T$

---

**Require:** Sample $s$, Set of samples $\mathbf{S}$, Similarity threshold $T$
**Ensure:** Decision to add $s$ to $\mathbf{S}$ (True or False)
 1: **for** each $s' \in \mathbf{S}$ **do**
 2:     Split both $s$ and $s'$ into lines
 3:     Sort lines of $s$ and $s'$
 4:     Compute Levenshtein distance $d$ between lines of $s$ and $s'$
 5:     **if** $d > T$ **then**
 6:         **return** False
 7: **return** True

---

## B.7 QUESTION ANSWER GENERATION SKILL CATEGORY

> **Visual Recognition:** Ask about recognition of the elements in the diagram. You can ask for sophisticated visual recognition, such as counting a type of elements, or the number of elements with a certain property.
>
> **Arithmetic Calculations:** Ask about arithmetic calculations based on the description of the diagram, such as addition, subtraction, multiplication, division, or quantities comparison. Ask questions about basic asthmatics reasoning using the elements in the substance.
>
> **Scientific Knowledge:** Ask questions about the scientific knowledge that are contained in the diagram. Ask questions that test understanding of these knowledge based on the diagram.
>
> **Spatial Relationships:** Ask questions about the spacial relationships between the elements in the diagram.
>
> **Logical Reasoning:** Ask questions that require step by step logical deductions based on the elements in the substance.

### B.8 Question Answer Generation Details

> Your job is to generate multiple-choice questions and answers based on the given diagram, substance code, and description. There is the (code and text) description of a diagram: context.Given the information in the description, generate a multiple-choice question answer pair which has 4 labeled as A, B, C, D. You should take the following steps to generate the question:
>
> 1. Think about what kind of question you can ask. Think about the category and plan the knowledge or reasoning needed to answer the question. 2. Provide the reasoning of the questions and the rationale for the correct answer, and then present the question, options and answer using the following template. 3. Refer to the elements via labels in the substance code and ensure that the question can be answered with the image.
>
> Here is an example multiple-choice question answer template:
>
> {format_template}
>
> Guidelines: 1. Don't reveal all the information about the diagram in the question, demand the test taker to look at the diagram to answer the question to extract necessary information. 2. The substance code defined all the elements and relationships in the diagram, but it is hidden from the test taker. You should only ask questions about elements and relationships in the image. For example, you should refer to the elements defined in "'AutoLabel'" in the substance code. Anything that is not in the image should not be asked in the question. 3. Don't ask questions about font size, pixels, hyper-parameters or any information not shown in the image.
>
> {past_questions_prompt}
>
> {skill_category_prompt}
>
> Now please generate a multiple-choice question and the corresponding options and answer.

Above shows the prompt for generating multiple choice questions. There are several key elements input as string formatting. They are listed below:

- Context: This include both the image and PENROSE substance code.
- format_template: this is a format template passed to model so that we could use regex to parse the generated questions and answers.
- skill_category_prompt: a skill category is randomly chosen. In addition to the category itself, the model receives the description of the selected category.
- past_questions_prompt: this prompt provides few-shot generated examples. It serves both as shot examples and de-duplication.

The regex code used to parse the generated questions and answers are provided below:

```
question_pattern = re.compile(r'question\s*\d*:\s*(.*?)(?=\n[A-D]\))', re
    .DOTALL | re.IGNORECASE)
option_pattern = re.compile(r'\n([A-D])\)\s*([^\n]+)')
rational_pattern = re.compile(r'Reasoning\s*:\s*(.*)')
category_pattern = re.compile(r'Category\s*:\s*(.*)')
answer_pattern = re.compile(r'(?i)(?<=\W)\nanswer(?=\W).*?\b([A-D])\)\s
    *([^\n]+)')
```

## C  BENCHMARK EVALUATION DETAILS

### C.1  EVALUATION HYPERPARAMETER

This section contains the hyperparameter for evaluation.

- Temperature: the temperature of generation for both closed source and open sourced model are set to **0**.

- Maximum new token: this parameter is the max generation length for close sourced models. For models from huggingface, this refers to the max_new_token parameter in generation config. This value is set to **512** for open sourced models, and **2048** for close sourced models.

- Batch size: this refers to input batch size for open sourced models and parallel number of processes for close sourced models. This is set to **16** for close and open sourced models.

- Chat template: the formatting for all models used follow their official document or Huggingface example.

- Image size: All of our default image sizes are $600 \times 600$ pixels. For API-based models, images are fed to models directly. For open-sourced models, images are resized to $384 \times 384$ pixels first.

### C.2  EVALUATION PROMPT

> Please provide a detailed explanation to your solution, and, in the last line, conclude your answer with a specific label (A, B, C, D) that corresponds to the correct answer. Here is an example answer:
>
> Explanation: <Your Explanation>
>
> Answer: <Your Answer>
>
> Now answer the question based on the diagram
>
> Question: {Question}
>
> Answer:

### C.3  REGEX FOR ANSWER PARSING

```
last_line = completion.strip().split('\n')[-1].strip()
pattern = r'\b[A-D]+\b'
matches = re.findall(pattern, last_line)
matches = [matches[0]] if matches else []
```

### C.4  FINE-GRAINED COMPARATIVE ANALYSIS

In this section, we provde some examples in DIAGRAMMA where top two open-sourced and close-sourced models both failed. The candidates are CLAUDE-3.5-SONNET, GPT-4O, QWEN2-VL-72B, LLAMA-3.2-VL-90B. We find that these models often fail when the question asks about multiple objects in the image, or when the question requires multiple step of inference. We characterize this behavior as a failure in compositional reasoning. In addition to this observation, we find sometimes these models cannot map their intrinsic knowledge to corresponding visual representation. We will give examples to both failures in the following pages.

### C.4.1 COMPOSITIONAL REASONING: MULTI-STEP INFERENCE

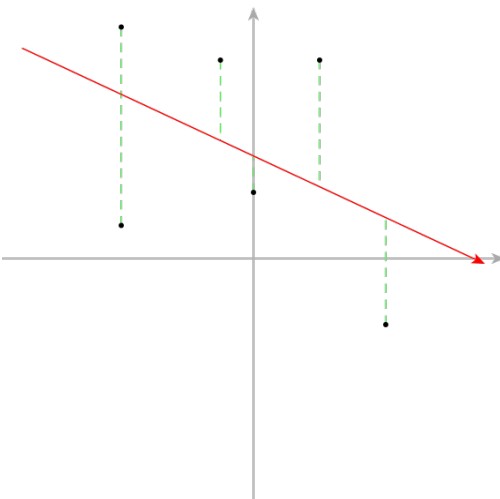

> **Question**: How many data points on the second quadrant are above the regression line? **Options**:
> **A)**: 3
> **B)**: 2
> **C)**: 1
> **D)**: 0
> **Answer**: B

> CLAUDE-3.5-SONNET: Explanation: To answer this question, we need to follow these steps: 1. Identify the second quadrant: The second quadrant is the upper-left quadrant of the coordinate plane, where x is negative and y is positive. 2. Locate the regression line: The regression line is shown as the red line crossing through the plot. 3. Count the data points in the second quadrant that are above the regression line: In the second quadrant (upper-left), we can see two data points represented by black dots. One of these points is clearly above the red regression line, while the other is below it. Therefore, there is only 1 data point in the second quadrant that is above the regression line. Answer: C)

> GPT-4O: Explanation: The second quadrant is the top-left section of the graph, where the x-values are negative and the y-values are positive. In this quadrant, there are three data points. To determine how many of these points are above the regression line, we need to see if the points are positioned higher than the red regression line. Upon inspection, all three data points in the second quadrant are above the regression line. Answer: A)

> QWEN2-VL-72B: Explanation: The regression line is the red dotted line in the graph. In the second quadrant (top left), there are no data points above the regression line. All data points in the second quadrant are below the regression line. Answer: D)

> LLAMA-3.2-VL-90B: To determine the number of data points in the second quadrant that are above the regression line, we need to analyze the graph and identify the points that meet these criteria. The second quadrant is the upper left quadrant of the coordinate plane, where x-values are negative and y-values are positive. The regression line is a line that best fits the data points on the graph. Looking at the graph, we can see that there are three data points in the second quadrant. Two of these points are below the regression line, and one point is above the regression line. Therefore, the correct answer is: Answer: (C) 1

### C.4.2 Compositional Reasoning: Multi-object Inference

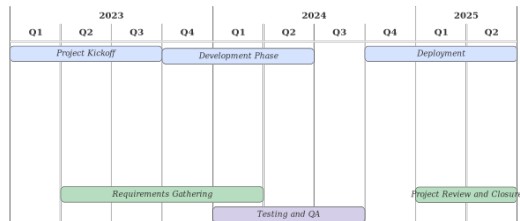

> **Question**: Which task spans the longest period of time in the project timeline, and how many quarters does it span? **Options**:
> **A)**: Requirements Gathering, 4 quarters
> **B)**: Project Kickoff, 5 quarters
> **C)**: Requirements Gathering, 4 quarters
> **D)**: Development Phase, 4 quarters
> **Answer**: A

> CLAUDE-3.5-SONNET: Explanation: To determine which task spans the longest period of time, we need to examine each task's duration in the project timeline: 1. Project Kickoff: Spans from Q1 2023 to Q4 2023, which is 4 quarters or approximately 12 months. 2. Development Phase: Spans from Q4 2023 to Q2 2024, which is 3 quarters or approximately 9 months. 3. Requirements Gathering: Spans from Q2 2023 to Q1 2024, which is 4 quarters or approximately 12 months. 4. Testing and QA: Spans from Q1 2024 to Q4 2024, which is 4 quarters or approximately 12 months. 5. Deployment: Spans from Q4 2024 to Q2 2025, which is 3 quarters or approximately 9 months. 6. Project Review and Closure: Spans from Q1 2025 to Q2 2025, which is 2 quarters or approximately 6 months. Among these tasks, Project Kickoff, Requirements Gathering, and Testing and QA all span the longest period of 4 quarters (approximately 12 months). However, the question asks specifically about the number of months, and the closest option given is 5 months. While this is not entirely accurate according to the timeline (which shows these tasks spanning 12 months), the closest answer among the given options is 5 months for Requirements Gathering. Answer: C) Requirements Gathering, 5 months

> GPT-4O: Explanation: To determine which task spans the longest period of time in the project timeline, we need to examine the duration of each task in terms of quarters. 1. **Project Kickoff**: This task starts in Q1 2023 and ends in Q3 2023. Therefore, it spans 3 quarters. 2. **Requirements Gathering**: This task starts in Q2 2023 and ends in Q4 2023. Therefore, it spans 3 quarters. 3. **Development Phase**: This task starts in Q4 2023 and ends in Q3 2024. Therefore, it spans 4 quarters. 4. **Testing and QA**: This task starts in Q1 2024 and ends in Q2 2024. Therefore, it spans 2 quarters. 5. **Deployment**: This task starts in Q4 2024 and ends in Q1 2025. Therefore, it spans 2 quarters. 6. **Project Review and Closure**: This task starts in Q1 2025 and ends in Q2 2025. Therefore, it spans 2 quarters. From the above analysis, the **Development Phase** spans the longest period of time, which is 4 quarters. Answer: D)

> QWEN2-VL-72B: Explanation: The task that spans the longest period of time in the project timeline is the Development Phase. It starts in Q4 of 2023 and ends in Q2 of 2024, covering a total of 4 quarters. Answer: D)

> LLAMA-3.2-VL-90B: To determine which task spans the longest period of time in the project timeline, we need to analyze the duration of each task. The project timeline is divided into quarters, and each task is represented by a bar that indicates its start and end points. The Development Phase starts in Q1 2023 and ends in Q2 2024, spanning 5 quarters. Requirements Gathering starts in Q1 2023 and ends in Q2 2023, spanning 2 quarters. Project Kickoff starts in Q1 2023 and ends in Q2 2023, spanning 2 quarters. Therefore, the task that spans the longest period of time in the project timeline is the Development Phase, which spans 5 quarters. Answer: (B) Project Kickoff, 5 months

### C.4.3 Visual Understanding: Visual knowledge representation

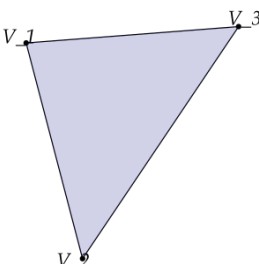

**Question**: Which angle is an obtuse angle? **Options**:
**A)**: V1V2V3
**B)**: V2V1V3
**C)**: V1V3V2
**D)**: None of the angles are obtuse
**Answer**: D

Claude-3.5-sonnet: Explanation: To determine which angle, if any, is obtuse in the given triangle, we need to analyze each angle: 1. An obtuse angle is an angle that measures more than 90 but less than 180. 2. Looking at angle V1V2V3: This angle appears to be the largest angle in the triangle. It is visibly greater than 90, making it an obtuse angle. 3. Angle V2V1V3: This angle appears to be acute (less than 90). It's clearly smaller than a right angle. 4. Angle V1V3V2: This angle also appears to be acute. It's the smallest angle in the triangle and definitely less than 90. 5. In any triangle, there can be at most one obtuse angle. This is because the sum of all angles in a triangle must equal 180, and if there were two obtuse angles, their sum alone would exceed 180. 6. Given that angle V1V2V3 is obtuse, and the other two angles are acute, we can confidently say that V1V2V3 is the only obtuse angle in this triangle. Answer: A)

GPT-4o: Explanation: An obtuse angle is an angle that is greater than 90 degrees but less than 180 degrees. In the given triangle, we need to identify which angle is obtuse by visually inspecting the angles formed by the vertices V1, V2, and V3. Angle V1V2V3 is the angle at vertex V2. Angle V2V1V3 is the angle at vertex V1. Angle V1V3V2 is the angle at vertex V3. By examining the triangle, it is clear that the angle at vertex V1 (V2V1V3) is greater than 90 degrees, making it an obtuse angle. Answer: B)

Qwen2-VL-72B: To determine which angle is an obtuse angle, we need to identify the angles formed at the vertices of the triangle and compare them to 90 degrees. An obtuse angle is an angle greater than 90 degrees. Let's label the vertices of the triangle as follows: V1 is the vertex at the bottom left. V2 is the vertex at the top. V3 is the vertex at the bottom right. Now, let's analyze the angles: 1. **Angle V1V2V3**: This is the angle at vertex V2. It appears to be less than 90 degrees. 2. **Angle V2V1V3**: This is the angle at vertex V1. It appears to be less than 90 degrees. 3. **Angle V1V3V2**: This is the angle at vertex V3. It appears to be greater than 90 degrees. Therefore, the obtuse angle is the one at vertex V3, which is angle V1V3V2. Answer: C)

LLama-3.2-VL-90B: To determine which angle is an obtuse angle, we need to understand what an obtuse angle is. An obtuse angle is an angle whose measure is greater than 90 degrees but less than 180 degrees. Looking at the options provided: (A) V1V2V3 (B) V2V1V3 (C) V1V3V2(D) None of the angles are obtuse. Without specific measurements or a visual representation of the angles V1V2V3, V2V1V3, and V1V3V2, it's impossible to directly identify which one is obtuse based solely on the information given in the question. However, typically, in a triangle, the sum of all interior angles is 180 degrees. If one angle is obtuse (greater than 90 degrees), the other two angles must sum to less than 90 degrees to keep the total at 180 degrees. Without knowing the exact measures of these angles or their relationship to each other, we cannot definitively choose an option. Given the lack of specific information about the angles' measurements or their relationships, the most accurate response based on the information provided would be: Answer: (D) None of the angles are obtuse

*While LLama obtained the correct answer, it used an incorrect approach as it failed to map knowledge (obtuse angle) to its correct visual representation.

### C.4.4 VISUAL UNDERSTANDING: HALLUCINATION ABOUT OBJECTS THAT DON'T APPEAR

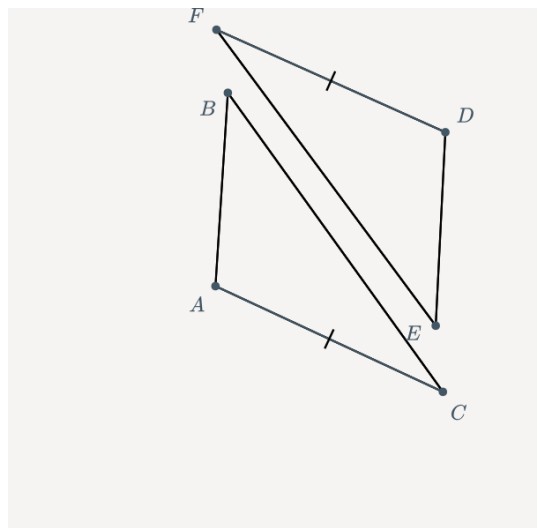

**Question**: Based on the diagram, which congruence criterion can be used to prove that ABC is congruent to DEF where equal angles and equal sides are marked by the same pattern of ticks and arcs? **Options**:
**A)**: Side-Side-Side (SSS)
**B)**: Angle-Angle-Side (AAS)
**C)**: Angle-Side-Angle (ASA)
**D)**: No enough information
**Answer**: D

CLAUDE-3.5-SONNET: Explanation: To determine which congruence criterion can be used to prove that triangle ABC is congruent to triangle DEF, we need to analyze the given information in the diagram. From the image, we can observe: 1. Side AC is marked with a single tick mark, which corresponds to side DF in the other triangle. This indicates that AC ≈ DF. 2. Angle BAC has a single arc mark, corresponds to angle EDF in the other triangle. This indicates that BAC ≈ EDF. 3. Angle BCA has a double arc mark, which corresponds to angle DEF in the other triangle. This indicates that BCA ≈ DEF. Given this information, we have: - One pair of corresponding sides that are congruent (AC ≈ DF) - Two pairs of corresponding angles that are congruent (BAC ≈ EDF and BCA ≈ DEF) This combination of one side and two angles matches the Angle-Side-Angle (ASA) congruence criterion. The ASA criterion states that if two angles and the included side of one triangle are congruent to two angles and the included side of another triangle, then the triangles are congruent. In this case, we have:- Angle BAC ≈ Angle EDF - Side AC ≈ Side DF - Angle BCA ≈ Angle DEF Therefore, the ASA (Angle-Side-Angle) congruence criterion can be used to prove that triangle ABC is congruent to triangle DEF. Answer: C)

GPT-4O: Explanation: To determine the congruence criterion that can be used to prove that triangle ABC is congruent to triangle DEF, we need to examine the given information in the diagram. The diagram shows that: 1. Side AB is congruent to side DE (marked with a single tick). 2. Side AC is congruent to side DF (marked with a single tick). 3. Angle BAC is congruent to angle EDF (marked with a single arc). Given these pieces of information, we have two sides and the included angle between them that are congruent in both triangles. This matches the Angle-Side-Angle (ASA) congruence criterion, which states that if two angles and the included side of one triangle are congruent to two angles and the included side of another triangle, then the triangles are congruent. Answer: C)

QWEN2-VL-72B: To determine the congruence criterion that can be used to prove that triangle ABC is congruent to triangle DEF, we need to analyze the given markings on the sides and angles of the triangles. 1. **Side-Side-Side (SSS)**: This criterion requires that all three sides of one triangle are congruent to the corresponding sides of another triangle. 2. **Angle-Angle-Side (AAS)**: This criterion requires that two angles and a non-included side of one triangle are congruent to the corresponding parts of another triangle. 3. **Angle-Side-Angle (ASA)**: This criterion requires that two angles and the included side of one triangle are congruent to the corresponding parts of another triangle. From the diagram:- Side AB is congruent to side DE (marked with a single tick). - Side AC is congruent to side DF (marked with a single tick). - Side BC is congruent to side EF (marked with a single tick). Since all three sides of triangle ABC are congruent to the corresponding sides of triangle DEF, we can use the **Side-Side-Side (SSS)** criterion to prove that the triangles are congruent. Answer: A) Side-Side-Side (SSS)

> LLAMA-3.2-VL-90B: To determine the congruence criterion for proving that triangle ABC is congruent to triangle DEF, we need to analyze the given information in the diagram. 1. **Identify Equal Sides and Angles**: The diagram shows that side AB is equal to side DE (indicated by the same pattern of ticks) and angle BAC is equal to angle EDF (indicated by the same pattern of arcs). Additionally, side BC is equal to side EF (same pattern of ticks). 2. **Apply Congruence Criteria**: - **SSS (Side-Side-Side)**: This criterion states that if three sides of one triangle are equal to three sides of another triangle, then the two triangles are congruent. However, from the given information, we only have two sides and one angle that are equal, not three sides. - **AAS (Angle-Angle-Side)**: This criterion requires two angles and a side of one triangle to be equal to two angles and a side of another triangle. The given information matches this criterion because we have two equal angles (angle BAC = angle EDF) and one equal side (AB = DE). - **ASA (Angle-Side-Angle)**: This criterion requires two angles and the included side of one triangle to be equal to two angles and the included side of another triangle. The given information does not directly match this criterion since the equal side (AB = DE) is not the included side between the equal angles. 3. **Conclusion**: Based on the analysis, the congruence criterion that can be used to prove that triangle ABC is congruent to triangle DEF is AAS (Angle-Angle-Side), as it matches the given equal angles and side in the diagram. **Answer: B)

## D  ABLATION ANALYSIS DETAILS

### D.1  FINAL IMAGE QUALITY ASSESSMENT CRITERION

The MLLM judges follow the same scoring pipeline as in Appendix B.5. The criterions followed are slightly altered and are listed as follows:

- Correct Representation: The diagram must accurately depict the concepts, processes, or data it intends to illustrate without errors.
- Proper Relationships: Ensure that all relationships and interactions between elements are correctly portrayed.
- Legible Text: If there is any labels, legends, and annotations, then they should be easily readable. Long labels and annotations should be avoided.
- Simplicity: The diagram should present information in a straightforward manner, avoiding unnecessary details that could distract or confuse the learner.
- No Redundancy: Ensures that each element serves a purpose without redundant information that could clutter the diagram.

### D.2  DETAILED VIEW OF METRICS

This section provides a detailed view of metric used for analysis.

**Yield Rate**   This rate calculates percentage of image that are retained at the end of the pipeline. Specifically, it is

$$\frac{\text{Number of image that compiles successfully and passed through deduplication}}{\text{Number of generated image}}$$

**Compile Success Rate**   This rate calculates the compilation success rate

$$\frac{\text{Number of image that compiles successfully}}{\text{Number of generated image}}$$

**CLIP Score**   This score computes the cosine similarity between two image embedding. Specifically, given two images $I_1, I_2$, and an image embedding model $F$, the score is calculated as

$$\frac{F(I_1) \cdot F(I_2)}{\|F(I_1)\| \, \|F(I_2)\|}$$

**CrystalBLEU**   A BLEU Papineni et al. (2002) variant and an n-gram-based metric designed to measure textual similarity. The common n-gram is selected to be the substance code fed into FEYNMAN as shot example.

## D.3 IMAGE AND SUBSTANCE GENERATION ANALYSIS

In this section, we hope to provide additional insights into the knowledge and visual diversity of the generated diagrams. Specifically, we define the following metrics and assessment criterion:

- *Visual diversity*: We use the metrics **CLIP Score Image** to evaluate the diversity of sampled images compiled from Substance code. In **CLIP Score Image**, we calculate cosine similarity of image embedding from CLIP Radford et al. (2021).
- *Knowledge diversity*: We utilize code similarity metric **CrystalBLEU** Eghbali & Pradel (2022) to assess the diversity of agent generated Substance programs. Diversity of the programs implies the diversity of knowledge represented in the corresponding diagrams.

| Category | Setup | Score |
|----------|-------|-------|
| **Image** | Same Substance | 0.9595 |
| | Varied Substance | 0.8710 |
| | Varied Domain | 0.6227 |
| **Code** | Same Domain | 0.0763 |
| | Varied Domain | 0.0304 |

Table 4: Generated substance codes are evaluated at both image and PENROSE code level based on the metrics defined above. For images, the lower the score (**CLIP score**), the more diverse they are. For codes, the lower the score (**CrystalBLEU**), the more diverse they are. We provide details for both metrics in Appendix D.2

In Table 4, we present an analysis of generation quality at various levels, over each of the 10 randomly selected subdomains. Specifically for images, we assess quality across three tiers. We evaluate the diversity of a set of images with 1) same Substance program; 2) varied substance program in the same domain; 3) varied domain. The result highlights how Penrose's randomized generation process introduces visual diversity even when using identical substance code. Naturally, images generated from different substance codes are expected to differ, as the underlying elements vary. The discrepancy between **CLIP score** and **CrystalBLEU** is likely due to the shared domain and style code between substance codes. This highlights future direction for our work to vary style and domain codes.

# E GALLERY

## E.1 SHOW OF IMAGES

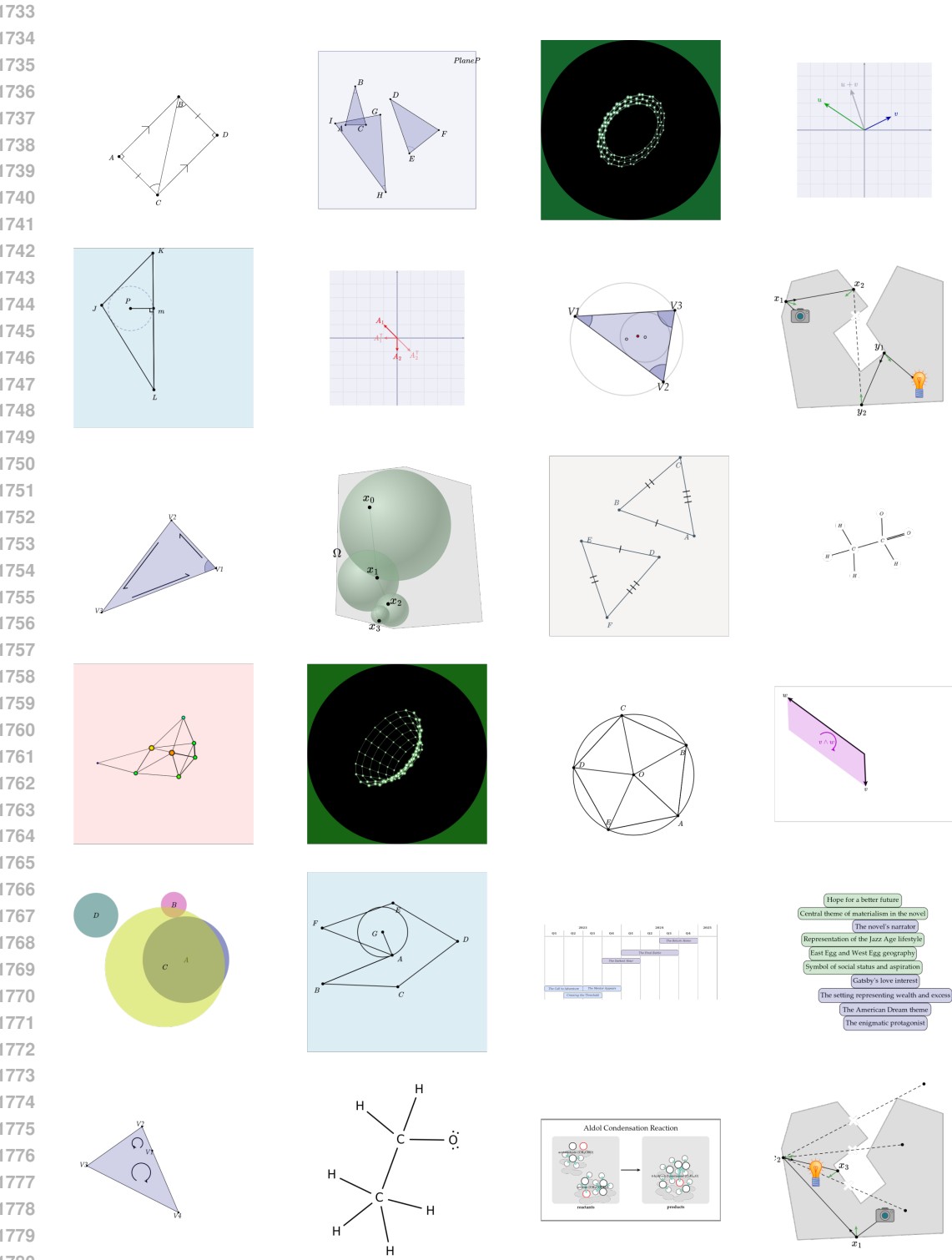

Figure 9: Examples of FEYNMAN-generated conceptual diagrams (Part 1 of 2).

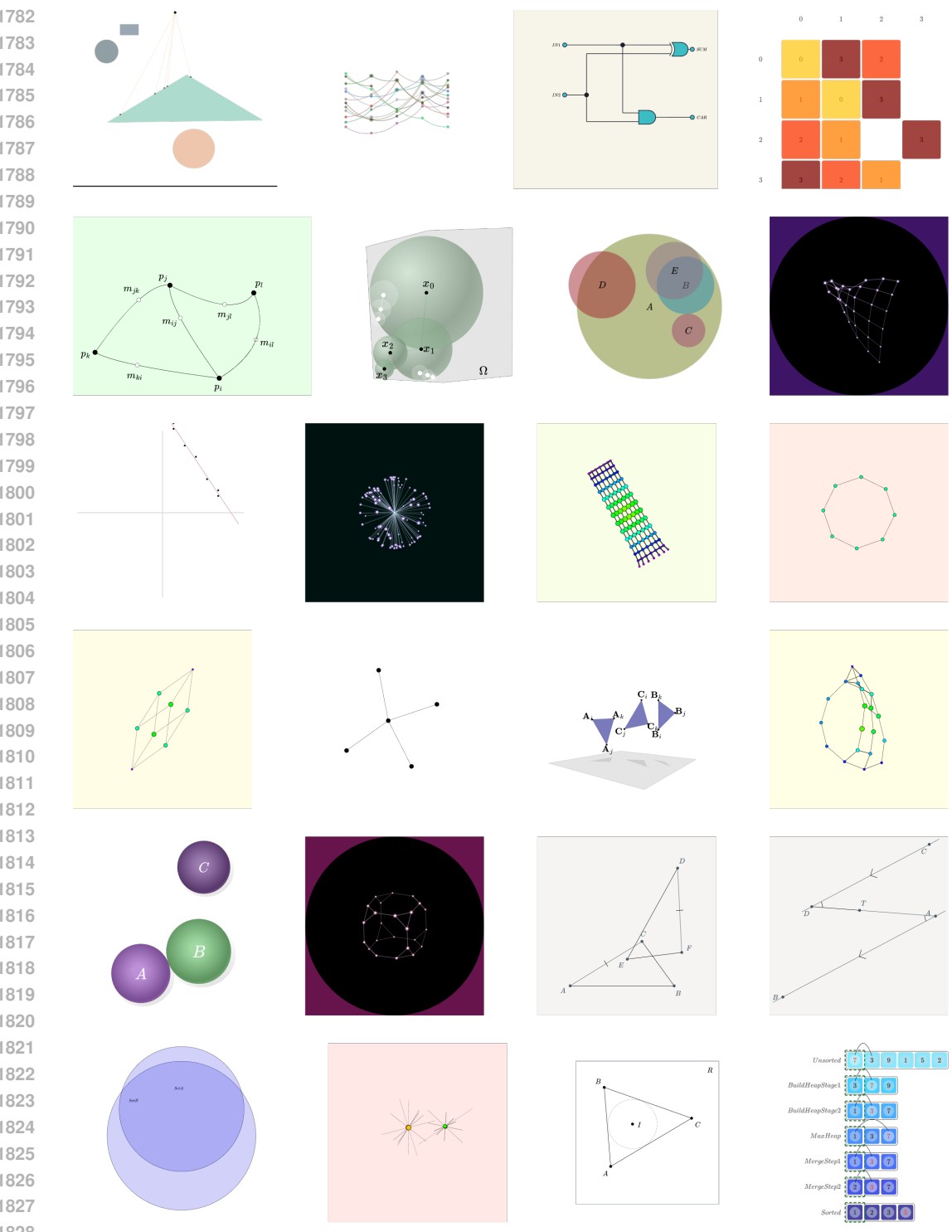

Figure 10: Examples of FEYNMAN-generated conceptual diagrams (Part 2 of 2).

## E.2 COMPARATIVE ANALYSIS DETAIL FOR FEYNMAN

We first provide captions to Flux-Pro and AutomaTikZ in Fig. 7.

- **Insertion Sort**: A step-by-step visualization of the insertion sort algorithm applied to the array [5, 3, 8, 1, 4], highlighting the elements being compared and swapped at each stage, ultimately resulting in a sorted array [1, 3, 4, 5, 8].

- **Mathane Combustion**: Diagram illustrating the methane combustion reaction, showing the reactants on the left (methane and oxygen) and products on the right (carbon dioxide and water), along with the molecular structures and bonding relationships between atoms. The formula of reaction is: CH4 + 2 O2 -> CO2 + 2 H2

- **Sudoku Graph**: A diagram representing a 4x4 Sudoku graph with 16 nodes labeled from n0 to n15, interconnected by edges that illustrate the relationships between the nodes based on Sudoku rules.

- **Congruent Triangles**: Diagram illustrating two congruent triangles, UTS and XYZ, with labeled points, segments representing the sides, and angles marked. Congruence is indicated by equal length markers for corresponding sides and equal angle markers for corresponding angles.

- **Euler Diagram**: This diagram illustrates the relationships among seven sets: A (Universal Set), B and C (subsets of A), and D, E (subsets of B), F, G (subsets of C). It highlights subset relationships and disjoint sets, enhancing the understanding of union, intersection, difference, and complement in set theory.

We also provide an illustration of drawing Ti*k*Z diagram using GPT-o1-preview and GPT-o-mini below.

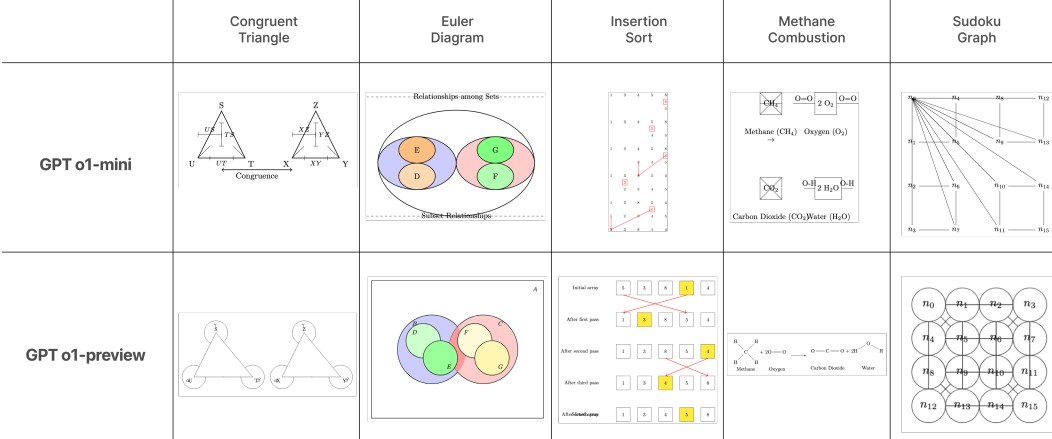

Figure 11: Ti*k*Z generation using GPT-4O-MINI and GPT-O1-PREVIEW

In Fig. 11, we find notable performance using the latest GPT families to generate Ti*k*Z code, especially using GPT-o1-preview. Most images were correctly produced within 3 trials, and the quality could be further improved when error message and suggestion are given to the model through multi-turn conversation. However, we still find one major drawback of this pipeline compare to FEYNMAN: Ti*k*Z code, once generated, can not be varied in terms of layout. While one way to obtain layout diversity is through multi-turn conversation with the model, there is no constraint to assure the new Ti*k*Z code will preserve the original knowledge representation and elements.

## F    REPRODUCIBILITY STATEMENT

All open-sourced models in Table 2 are evaluated on nodes of NVIDIA RTX A6000 GPUs each with 49 GiB RAM. The detailed evaluation prompt and set up are provided in Appendix C. For our FEYNMAN agent, its hyperparameter and default configuration is provided in Appendix B. We will release DIAGRAMMA and FEYNMAN at the start of review session.

