# OpenReview forum: "Feynman: Knowledge-Infused Diagramming Agent for Scaling Visual Reasoning Data"
_ICLR.cc/2025/Conference — Submitted to ICLR 2025_

### Official Review · Reviewer_JbwN · 2024-10-27

**Soundness:** 2
**Presentation:** 2
**Contribution:** 2
**Rating:** 3
**Confidence:** 4

**Summary:**

The authors introduce Feynman, a diagram-generation agent.  Feynman leverages Penrose, a diagramming tool that generates vector graphic diagrams from its own programming codes. Feynman’s diagram synthesis has four steps: idea, plan, iterate, and render.

- In idea step, an LLM is given a text description of target diagram and generates relevant ideas/concepts in plain text.
- In plan step, an LLM is given the documentation of Penrose and in-context examples and generates visual elements.
- In iterate step, a panel of visual judges (MLLMs) iteratively provide the feedback and the generation plan gets refined.
- Render step is not explicitly described, but perhaps Penrose tool would render the iteratively refined plan from the previous step into a diagram.

The authors also provide a benchmark called Diagramma, which contains 1058 multiple-choice QAs on the diagrams generated with Feynman.

The authors provide a brief qualitative comparison of Feynman and baselines (AutomaTikz and Flux-Pro). They also provide the evaluation of different MLLMs on Diagramm, and ablation of different planning components within Feynman.

**Strengths:**

- Introduction of Feynman, an LLM-based diagram generation framework that uses Penrose as a backend.
- Introduction Diagramma, a VQA dataset to evaluate  diagram understanding capability of MLLMs where the diagram images are generated with Feynman

**Weaknesses:**

- **Unclear discussion about Penrose vs. TikZ.** In the introduction, the authors mention that “AutomaTikz still exhibits efficiency overhead in synthesizing scientific diagrams at scale, due to the inherent complexity of both the TikZ language and visual design.” However, the authors do not compare using TikZ or Penrose as a rendering backbone. There must be different scenarios when Tikz and Penrose are more useful, and the authors should discuss them.
- **Weak comparison with baselines.** The authors only provide brief qualitative examples in Fig. 7, but it is unclear how the five input prompts are selected. In addition, compared to AutomaTikz, Feynman uses stronger LM (GPT-4) than Llama 2 used in AutomaTikz. This makes it hard to understand the fundamental advantage of Feynman compared to baselines such as AutomaTikz.
- **Missing related work.** DiagrammerGPT (COLM 2024; https://arxiv.org/abs/2310.12128) seems to be highly relevant, as it also uses an LLM for diagram layout planning (w/ iterative refinement) and uses diffusion or different vector graphic tools (e.g., PowerPoint/Inkscape/illustrator) for rendering. They also provide a training annotation and benchmark in the scientific domain - AI2D caption. I highly encourage authors to add discussion and comparison to this work.
- **Weak analysis of the correctness of the generated diagrams.** The highest performance among its ablation study (in Table 5) does not mean that the generated diagrams are correct. The authors should provide a more rigorous analysis of the correctness of the diagrams generated by Feynman (e.g., human evaluation). It is unclear whether Diagramma can be a meaningful evaluation benchmark of MLLMs without such analysis.

**Questions:**

- The Penrose diagram tool should be cited (with URL) the first time it’s mentioned.
- In Sec. 2, the subsection names do not directly match with the four steps mentioned in the first paragraph of Sec 2: idea, plan, iterate, and render. For example, Sec 2.2 is ‘Knowledge Planning' instead of ‘idea'; there's no subsection for the last step - rendering. Explicitly matching the subsection names with the four steps would make the readers easily match which subsection describes which step.
- Is Sec 2.5 part of the ‘iterate’ step? it is not clear how the QA data is generated and used. The step needs to be explained in more detail.

---

> ### Author Response · Authors · 2024-12-03
>
> Thank you for taking time to read our paper and the meaningful feedback you provided. Below we address some of the issues raised.
>
> ---
> **Unclear discussion about Penrose vs. TikZ**
>
> Thank you for pointing this out. There are some general tradeoffs between using TikZ and Penrose. TikZ offers granular control suitable for users who need detailed customization and are comfortable with manual specification. While Penrose provides a higher-level approach, automating the visualization process by interpreting the relationships between objects, which can be advantageous for quickly generating diagrams without delving into graphical details.
>
> *In our use case, Penrose helps with scalability*. Penrose code decoupled the manipulation of low-level graphical details and high-level concepts, and LLM are better at the high-level reasoning and knowledge extraction, which we try to utilize. TikZ is potentially more powerful if the data size continues to increase with more TikZ diagrams on the internet. However, different from textual math or code data, TikZ code and diagrams usually are less readable and have worse captions, limiting their quality and usability.
>
> ---
>
> **Weak comparison with baselines.**
>
> We believe that our choice of using the Llama-2 model released by AutomaTikZ paper is a suitable choice as baseline. In the work of AutomaTikz, they compared with GPT-4 baseline and claimed that their trained LLama-2 + MCTS produce better results, which is why we base our comparison on their Llama-2 model. They also explained in their paper why the current LLM API does not support their approach of using MCTS for code generation. We took the code and model from their implementation and tested against our results, as the reproduction of their work based on Llama-3 is beyond our capability.
>
> In the appendix, we also compared the non-agent approach that uses a much stronger coding model such as OpenAI o1-preview to generate TikZ code. We found that even though o1 is very good at writing TikZ code that can be successfully compiled into diagrams, its output diversity is rather limited, possibly due to the inherent difficulty of diversifying the visual layout in TikZ code.
>
> ---
> **Missing related work of DiagrammerGPT**
>
> Thank you for pointing out this extremely relevant work that we were not aware of. DiagrammerGPT is an agentic model that generates a text plan with LLM and uses a diffusion model for diagram generation. It shares a similar goal to our work and is pretty advanced in terms of their techniques and results. However, we realize there are some major differences between our approaches, which makes our approach potentially better in some domains.
> 1. The language planning part of diagram generation in DiagrammerGPT is very similar to our approach. The major distinction between our approach and theirs is that our code planning step serves to improve the following coding process (which could be done by either a fine tuned or off-the-shelf LLM), while they trained a diffusion model to read the plan as input.
> 2. The image generation part has similar drawbacks to the approach of using diffusion models directly, where the elements drawn by the diffusion models contain rich visual features that are beyond the conceptual relations described in the plan. We think their image generation is more suitable for illustration of scientific knowledge where natural image generation is helpful. While our approach of using programming tools is better for abstract diagramming due to the clarity of conceptual programming.
> 3. We believe that our approach is more suitable for scaling the complexity of abstract/geometric relationships. Here the ability of generating complex diagrams depends on the coding and reasoning ability of LLMs, which is much easier to scale and improve than the diffusion models.
>
> ---
>
> **Weak analysis of the correctness of the generated diagrams.**
>
> We acknowledge the importance of conducting a more rigorous evaluation on the correctness of the diagrams generated by Feynman. But we also believe that there is some misconception about how we obtain the Benchmark Diagramma, which led to the unfair accusation that Feynman’s generations are of low quality.
> - **Diagramma is a benchmark curated by the authors manually**. We checked all the question-answer pairs, and we also filtered out the visually incoherent/unrecognizable diagrams, to improve benchmark quality. Although we cannot guarantee 100% correctness, we believe that the erroneous examples are sparse, since the majority of the image QA samples in the dataset are easy for humans (both visually and textually).
>
> - Since we have manually curated the benchmark dataset, we believe that the inability of MLMs to answer the questions in Diagramma is not due to low data quality, but something else. In fact, **we conjecture that our dataset is comparably more out-of-distribution** than most of the existing visual-language benchmarks for MLLMs.

---

### Official Review · Reviewer_AiEM · 2024-11-03

**Soundness:** 3
**Presentation:** 3
**Contribution:** 3
**Rating:** 6
**Confidence:** 3

**Summary:**

This submission presents FEYNMAN, a knowledge-infused diagramming agent designed to generate high-quality, well-aligned diagram-caption pairs at scale. ​ The agent decouples knowledge elicitation from visual production, leveraging LLMs for domain-specific knowledge enumeration and the PENROSE diagramming system for rendering. ​ The authors synthesized a dataset of over 100k diagram-caption pairs and curated a visual-language benchmark, DIAGRAMMA, to evaluate the visual reasoning capabilities of vision-language models.

**Strengths:**

There are several noticeable strengths in this paper:
* This paper presents a well-justified motivation for introducing LLMs into the pipeline of automatic diagram generation process.
* The manuscript is commendable for its clarity and structured writing style, which greatly facilitates reader comprehension.
* Additionally, the inclusion of clear and illustrative figures and tables is a notable strength, as it significantly aids in conveying the main claims of the paper to the audience.

**Weaknesses:**

Since this submission is one of the first studies on MLLM-assisted diagram generation agents, it could offer even greater value to future research if the authors provided a more comprehensive error analysis. Such analysis would help identify bottlenecks in the current pipeline and suggest directions for improvement.

Specifically, in the four steps of the FEYNMAN pipeline—idea, plan, iterate, and render—what types of errors occur? How are these errors distributed across the steps, and what might be their underlying causes? For example, are certain errors attributable to the limitations of the PENROSE program, or are others due to knowledge gaps or inaccuracies inherent in the LLMs? A detailed breakdown of these errors would provide valuable insights for future work.

**Questions:**

Please refer to the Weakness section.

---

> ### Author Response · Authors · 2024-12-03
>
> Thank you so much for your time reading our manuscript. Your suggestions on conducting a more detailed error analysis of the agent pipeline makes a lot of sense!
>
> Although we were not able to do such analysis systematically, we provide a qualitative view based on our running logs on some notable errors.
>
> ---
> **Idea**
> In this stage, the primary goal is to elicit knowledge of an LLM in a specific domain. For example, to draw structural chemical formulas, we ask FEYNMAN (GPT-4o backend) to enumerate n number of chemical reaction formulas. Something we observe is that when the model does not have ample knowledge in a particular domain, the given ideas have some repetition when n is large. We think that this is the reason caused discrepancies between scaling experiment shown in Fig. 8
>
> ---
> **Render/Iterate**
>
> We identify two types of errors at this stage:
> 1. Compile Errors: These occur when the model fails to generate correct Penrose code. We observed that providing compiler error messages as feedback significantly improved the model's code generation in subsequent rounds.
> 2. Diagram Errors: These arise when the code compiles successfully, but the generated diagram is flawed (e.g., overlapping shapes, illegible text). To address this, we utilized VLM judges with a critical mechanism to evaluate and refine these elements (details in Section B.5).
>
> Overall, both error types demonstrate that multi-round refinement with targeted critic feedback improves the alignment and quality of diagrams generated based on an inquiry.
>
> If you are interested, you are welcome to review the running log of agent pipeline yourself and the link to Diagramma, is provided here [https://anonymous.4open.science/r/feynman_anonymous-D5A7/README.md]

---

### Official Review · Reviewer_rjPm · 2024-11-03

**Soundness:** 2
**Presentation:** 3
**Contribution:** 2
**Rating:** 3
**Confidence:** 5

**Summary:**

This paper presented a scalable data generation pipeline with a diagramming agent, FEYNMAN. Through four steps, idea, plan, iterate and render, FEYNMAN decouples knowledge elicitation and visual production to create programs, which are then rendered into different layout diagram variations using penrose.


FEYNMAN generated 10693 programs, further resulting in 106,930 diagram-caption pairs. Additionally, it curated a scientific benchmark, DIAGRAMMA,  which contains 1,058 multiple-choice questions focused on visual understanding and reasoning.

The paper experimented with the generation capabilities of FEYNMAN pipeline across different subjects and the Appendix D.3 further discusses the diversity between programs and images under identical and distinct substances or domains. Furthermore, the paper evaluated DIAGRAMMA on 17 different MLLMs, confirming its validity for measuring the scientific visual reasoning capabilities of MLLMs.


Additionally, it qualitatively compared different diagramming approaches, discussing the relationship between the number of generate images and the total input/output tokens. At the same time, an ablation study was conducted to examine the impact of various pipeline components on the accuracy and quality of the generated programs.

**Strengths:**

- The paper conducted comprehensive experiments and analyses.

    - It studied the generation capabilities of the pipeline across different subjects, evaluated the performance of 17 MLLMs on DIAGRAMMA.

    - Additionally, compared FEYNMAN with other methods for generating knowledge-rich images, and  an ablation study was performed to examine the impact of different pipeline components.

     - The appendix provides detailed prompt designs for each component of the pipeline.

- The paper uses iterative visual refinement to obtain programs that better represent knowledge.

    - During the iterations, VLMs act as judges, providing visual judgments and feedback to refine the program and produce better images with correct representation, proper relationships, legible text, simplicity, no redundancy and other essential qualities.

**Weaknesses:**

- How many data were generated exactly? Is this in Section 3.1 mentions generating 10,693 unique substance programs across different subjects through the pipeline?

- To demonstrate the scalability of the pipeline, Section 4.2 discusses the relationship between the total input/output tokens and the number of generated images after de-duplication.  I don't quite understand the significance of this figure. The more samples are generated, the more tokens are naturally consumed. This seems like correct but useless talk.

- In Section 4.3, ablation experiments were conducted on 10 subdomains, but the experimental conditions and analysis are somewhat vague.
    - What exactly is the specific experimental design? How are these several pipelines compared?
    - What is the specific number of samples in each sub-domain?
    - How exactly were KP, CP, and S ablated? If CP is missing, how is code generation achieved?
    - In Table 3, Why does the Yield and Compile of CP + S differ by about 10% when KP is absent?
    - Were these evaluation metrics designed by the author (PR, LR,...) all given by VLMs? Is this accurate enough?

- In the iterative process, feedback is provided by GPT-4o, Claude-3.5-sonnet, and Gemini-Pro-1.5. However, what I want to say is that this evaluation seems to be inaccurate. The benchmark generated in this paper evaluates the performance of VLMs on such math-related charts. Judging from Table 2, the highest performance is only 60%. It seems unreliable to use these VLMs for evaluation and provide feedback.

**Questions:**

As I mentioned above.

- What exactly is the significance of Figure 8?
    - Can the relationship between tokens and the number of programs demonstrate scalability?
    - There are too few data points shown in the figure. It seems that it is difficult to observe obvious patterns when fitting these lines.
    - It shows the relationship between total input/output tokens and the number of generated images after de-duplication. Does it consider the number of tokens consumed in the iterative process?

- In Algorithm 1
    - Does setting different rendering seeds affect the scoring?
    - Is it a binary score (good/bad), or is it a score given within a specific range?

- If we remove Iterative Visual-Refine based on VLMs and only reflect based on feedback by the compiler. What will be the effect? How many samples can it generate?

- Please elaborate in more detail on what the innovation of the paper is? It seems that generating charts through code as a medium has been explored by some works.

- FEYNMAN generated over 106,930 diagram-caption pairs, yet DIAGRAMMA only contains 1,058 multiple-choice questions？ How many testing set in total?

- Could you provide a complete example, including each step of the pipeline’s output and the iteration process?

- Since no anonymous open-source link has been provided, the quality of the data cannot be determined. Hope to see the provision of an anonymous link in the subsequent rebuttal.

---

> ### Author Response · Authors · 2024-12-03
>
> Thank you for your time writing the review and reading the paper. We want to address some of your major concerns and questions.
>
> ---
> **How many data were generated exactly?**
>
> It seems like there is some confusion about our approach. We clarify here but please refer to Section 2 and the appendix in the paper for comprehensive description.
>
> As mentioned in the contribution, our pipeline generates caption and programs that compiles to image. There are ~110,000 of them.
> To further test the quality of our pipeline, we curated a benchmark, Diagramma, in which there are 1,058 question and answer pairs in total. Although the source data of the benchmark was created using LLMs, all 1,058 question-answer pairs in Diagramma have been manually verified for accuracy. Please refer to Table. 1 in the paper for details.
>
> ---
>
> **Significant of Section 4.2 and Figure. 8**
>
> It is correct of what you wrote about that token usage naturally increases as the number of samples grows. However, as we discuss in this section and highlight in the caption under Figure 8, the key observation is not the increasing trend itself but the curvature of the token usage. A linear trend suggests potential for further scaling within a specific domain as token counts continue to grow, whereas a decaying trend indicates diminishing returns in image generation as token usage increases.
>
> **The goal of this section is to demonstrate that, in certain domains, the number of unique generated images can be scaled up after deduplication, even as the number of samples rises.**
>
> ---
> **How exactly were KP, CP, and S ablated? If CP is missing, how is code generation achieved?**
>
> Each component refers to a distinct aspect of the prompting process. In your example, if the Code Planning (CP) component is omitted, the agent is instructed to directly write code for generating a Penrose program without employing a chain-of-thought step to plan the code generation.
>
> ---
>
> **Were these evaluation metrics designed by the author (PR, LR,...) all given by VLMs? Is this accurate enough?**
> We acknowledge that concerns may arise regarding the validity of using multiple LLM models to critique work generated by another LLM. However, we emphasize that the judges' evaluation primarily focuses on structural aspects of the image. For instance, determining whether an image contains legible text—one of the evaluation criteria—should be relatively straightforward and objective. Furthermore, it is some compromise we have to commit given the generation is at scale.
>
> ---
>
> **If we remove Iterative Visual-Refine based on VLMs and only reflect based on feedback by the compiler. What will be the effect? How many samples can it generate?**
>
> Thank you for this question. We have incorporated the results in the paper. As shown in Algorithm 1, VLM judges are only utilized when the compilation is successful. This means that feedback from the compiler serves as a baseline for the agent to produce correct programs. Consequently, there are two distinct forms of improvement:
> 1. **Improvements in program correctness**: Ensuring that the generated programs compile successfully.
> 2. **Improvements in image quality**: Enhancing the visual output of the generated programs.
> While rerunning the full scale-up experiment is resource-intensive, it is important to note that the compilation rate is not influenced by the inclusion of VLM judges. For insights into different setups, please refer to Table 3.
>
> ---
>
> **Please elaborate in more detail on what the innovation of the paper is? It seems that generating charts through code as a medium has been explored by some works.**
>
> Thank you for this question! We would like to highlight the following key innovations:
> 1. **Iterative improvement through visual judges and compilation feedback**: This approach enables the generation of correct and high-quality code programs by iteratively refining the outputs based on visual and compilation feedback.
> 2. **Conceptual diagramming**: Conceptual diagramming separates the visual production of layouts from the conceptual relationships of visual elements, which is a key ingredient of our approach and it help utilize LLM's knowledge capacity and improve the precision and clarity of generated contents. By leveraging *Penrose* as the backend for our diagramming tool, we were able to make the model enumerate its rich domain knowledge without the need to let LLM to fully control the layout process. Moreover, conceptual diagramming provides **knowledge diversity/flexibility** compared to fixed programmatic approach such as using matplotlib to draw flow charts.
>
>
> We hope that you recognize our work as an attempt to generate scientific diagrams at scale. To further address your concern, we provide following end results for our work in link provided here [https://anonymous.4open.science/r/feynman_anonymous-D5A7/README.md] It contains: the Diagramma benchmark 2. Generated diagram-caption  3.A few example illustrating the pipeline.

---

### Official Review · Reviewer_Tw8f · 2024-11-03

**Soundness:** 2
**Presentation:** 3
**Contribution:** 2
**Rating:** 3
**Confidence:** 4

**Summary:**

This work proposes the Feyman framework, which is tackling the challenges of scalable generations of knowledge-grounded diagrams. Feyman follows a code-generation and diversification fashion. At first, an LLM is used to propose some “ideas”, and then these ideas will be translated into codes following the Penrose template. The Penrose code can be formulated as a constraint optimization problem to derive diagrams. Then equipped with a visual feedback mechanism, Feyman generates visual diagrams with the codes and refine upon satisfaction. Eventually Feyman is used to generate knowledge-grounded visual QAs.
The authors then conduct some extensive studies on their synthetically generated data with various open-source and proprietary visual-language models.

**Strengths:**

- The whole approach of the proposed Feyman is sound.
- The work is tackling a challenging domain and indeed requires a much better scalable solution.
- The feedback mechanism and deduplication is interesting.

**Weaknesses:**

- More detailed statistics are needed for the proposed dataset. It is clueless to judge its quality, complexity, as well as how it would contribute/drive the research community.
- While Table 2 is much appreciated (for its comprehensiveness), what is the current plausible upper bound? Of course 100% correctness would be the ultimate ideal performance, but what is the realistic upper bound for the community to judge the progress?
- More details of the visual judges are needed. For example, what exactly are they? Are they fine-tuned, are they evaluated with a certain type of performance measurement? How do we know if the judges are faithful enough?

**Questions:**

- What are the visual judges?
- Why were there no human evaluations even at the very least for sanity checking conducted?
- Why choose the QA format after all the hard work? In addition, how do we envision the community would benefit from this dataset or any of the proposed methods?

---

> ### Author Response · Authors · 2024-12-03
>
> Thank you for your time writing the review and reading the paper. We want to address some of your major concerns and questions. And we also want to emphasize that the major goal of this paper is not to introduce a new benchmark but to demonstrate a scalable method for generating scientific diagrams with well-aligned text captions.
>
> ---
> **More details of the visual judges are needed, What are the visual judges?**
>
> *Most of the details are talked about in the paper.* We adopted open Visual Language Models (VLMs) as the judges, prompted to give binary scores on certain predefined criterions. Details about which model and prompts we used are provided in section B.5 in appendix. Our goal is mainly to showcase the possibility of iterative improvements from the feedback of visual judges, but there could be a significant room left for fine tuned judges in special domains.
> We acknowledge that concerns may arise regarding the validity of using multiple LLM models to critique work generated by another LLM. But this choice of our pipeline does have certain advantages.
> 1. It is more scalable (to dataset size) than human evaluation. To synthesize data at scale, common approaches such as human verification or programmatic verification fall short because of the diversity and knowledge coverage of the data generated. This is the major reason why we chose to use VLMs to provide feedback in a scalable manner.
> 2. It is possible to train customized VLMs to serve as judges. However, that requires VLM annotations which is also lacking currently and costly to obtain. We hope that our work could be a step towards enriching the data of scientific diagrams for various purposes.
> 3. We also want to emphasize that the judges' evaluation primarily focuses on structural aspects of the image. For instance, determining whether an image contains legible text—one of the evaluation criteria—should be relatively straightforward. Moreover, we expect these results to improve along with the other improvements in VLMs.
>
> ---
>
> **Why were there no human evaluations even at the very least for sanity checking conducted?**
>
> We would like to clarify that the benchmark Diagramma is human curated, where we have manually checked and corrected all mistakes spotted.
> For your reference, we have made both Diagramma and the image-text pairs publicly available here: [https://anonymous.4open.science/r/feynman_anonymous-D5A7/README.md]
>
> If the question is referring to the whole dataset we generated, we have several reasons for our failure to do so.
> 1. As all synthetic data generated by LLM or any neural networks, they are not error-free and are subject to the error patterns they learned in training. It is beyond the scope of this paper to check all the possible patterns incurred by the agent in generating the data, which involves significant manual labor.
> 2. We believe that the quality of the data will improve similarly to how math data quality in textual form gets improved, as we mainly used off-the-shelf LLM API for our agent pipeline. This relieves us with the task of improving the custom VLMs or LLMs to catch up with the progress every several months. In fact, our agent pipeline depends mainly on the LLM’s ability to code and enumerate domain knowledge and VLM’s visual recognition capability. It can be viewed or understood as some form of “self-distillation”, since we are trying to transform LLM’s textual knowledge into visual-language data.
>
> ---
>
> **Why choose the QA format after all the hard work?**
>
> Thank you for this question. QA format is just a choice we made as it is the mostly used format in visual instruction-tuning. The choice of the textual format of the image-text data is actually very flexible, as long as they are about the content of the images. Our approach naturally generates both diagrams and their corresponding code pairs by creating programming language representations of diagrams. We let LLM to summarize/describe the generated code, which is somewhat close to natural language, to obtain the text captions.
>
> ---
>
> **In addition, how do we envision the community would benefit from this dataset or any of the proposed methods?**
>
> We believe that the primary contribution, which is the **Feynman** pipeline, offers substantial value by enabling a scalable method for generating training pairs, particularly with images that are out-of-distribution compared to internet-sourced datasets.
>  **It is important to note that our primary objective is not to introduce a new benchmark but to demonstrate a scalable method for generating scientific diagrams.**

---

### Meta-Review · Area_Chair_98pA · 2024-12-20

**Metareview:**

This paper presents Feynman, a knowledge-infused diagramming agent for generating scientific diagrams and aligned captions at scale. While the approach of using LLMs for knowledge extraction and Penrose for rendering is interesting, several major concerns were raised by reviewers: (1) Insufficient evaluation of data quality and lack of human verification beyond the curated benchmark (Tw8f, rjPm), (2) Limited analysis of error patterns and bottlenecks in the pipeline (AiEM), (3) Weak baseline comparisons and missing discussion of key related work like DiagrammerGPT (JbwN). The primary contribution appears to be the pipeline itself rather than the benchmark, but more rigorous validation of the generated outputs is needed. Given these limitations in evaluation and analysis, I recommend rejection.

**Additional Comments On Reviewer Discussion:**

During the rebuttal period, the authors clarified several points: they emphasized that Diagramma (1,058 QA pairs) was manually verified despite being sourced from LLM generations, explained their choice of VLM judges for scalability, and provided an anonymous link to examine the data quality. However, reviewers remained concerned about the lack of systematic error analysis and human evaluation of the full 100K+ generated pairs. While the authors' responses addressed some technical clarifications, they did not fully resolve the core concerns about rigorous validation of the pipeline's outputs and comparison to state-of-the-art baselines.

---

### Decision · Program_Chairs · 2025-01-22

Reject